



# Exploring Constraints on a Wetland Methane Emission Ensemble (WetCHARTs) using GOSAT Satellite Observations

Robert J. Parker[1,2], Chris Wilson[3,4], A. Anthony Bloom[5], Edward Comyn-Platt[6,7], Garry Hayman[7], Joe McNorton[6], Hartmut Boesch[1,2], and Martyn P. Chipperfield[3,4]

[1]National Centre for Earth Observation, University of Leicester, UK
[2]Earth Observation Science, School of Physics and Astronomy, University of Leicester, UK
[3]National Centre for Earth Observation, University of Leeds, UK
[4]School of Earth and Environment, University of Leeds, UK
[5]Jet Propulsion Laboratory, California Institute of Technology, Pasadena, CA, USA
[6]European Centre for Medium-Range Weather Forecasts, Reading, UK
[7]Centre for Ecology and Hydrology, Wallingford, UK

**Correspondence:** R. J. Parker (rjp23@le.ac.uk)

**Abstract.**

Wetland emissions contribute the largest uncertainties to the current global atmospheric $CH_4$ budget and how these emissions will change under future climate scenarios is also still poorly understood. Bloom et al. (2017b) developed WetCHARTs, a simple, data-driven, ensemble-based model that produces estimates of $CH_4$ wetland emissions constrained by observations of precipitation and temperature. This study performs the first detailed global and regional evaluation of the WetCHARTs $CH_4$ emission model ensemble against 9 years of high-quality, validated atmospheric $CH_4$ observations from the GOSAT satellite. A 3-D chemical transport model is used to estimate atmospheric $CH_4$ mixing ratios based on the WetCHARTs emissions and other sources.

Across all years and all ensemble members, the observed global seasonal cycle amplitude is typically underestimated by WetCHARTs by -7.4 ppb, but the correlation coefficient of 0.83 shows that the seasonality is well-produced at a global scale. The Southern Hemisphere has less of a bias (-1.9 ppb) than the Northern Hemisphere (-9.3 ppb) and our findings show that it is typically the North Tropics where this bias is worst (-11.9 ppb).

We find that WetCHARTs generally performs well in reproducing the observed wetland $CH_4$ seasonal cycle for the majority of wetland regions although, for some regions, regardless of the ensemble configuration, WetCHARTs does not well-reproduce the observed seasonal cycle. In order to investigate this, we performed detailed analysis of some of the more challenging exemplar regions (Paraná River, Congo, Sudd and Yucatán). Our results show that certain ensemble members are more suited to specific regions, either due to deficiencies in the underlying data driving the model or complexities in representing the processes involved. In particular, incorrect definition of the wetland extent is found to be the most common reason for the discrepancy between the modelled and observed $CH_4$ concentrations. The remaining driving data (i.e. heterotrophic respiration and temperature) are shown to also contribute to the mismatch to observations, with the details differing on a region-by-region basis but generally showing that some degree of temperature dependency is better than none.



We conclude that the data-driven approach used by WetCHARTs is well-suited to produce a benchmark ensemble dataset against which to evaluate more complex process-based land surface models that explicitly model the hydrological behaviour of these complex wetland regions.

## 1 Introduction

The uncertainty in the emissions from natural wetlands remains the most significant uncertainty in the global $CH_4$ budget (Melton et al., 2013; Kirschke et al., 2013; Saunois et al., 2020). Without a more complete understanding of the processes by which wetlands emit $CH_4$, and, more importantly, how sensitive these processes are to changes in different driving factors such as precipitation, respiration and temperature, any attempt to model global $CH_4$ emissions for future climate change scenarios remains challenging and prone to large uncertainties (Zhang et al., 2017b; Ganesan et al., 2019). Previous studies (Bohn et al.,
2015; Poulter et al., 2017) have suggested that it is the uncertainty around wetland extent that is the largest contributor to uncertainties in the total methane emissions, with uncertainties in the climate response driving the interannual variability.

It is acknowledged (Kleinen et al., 2012; Stocker et al., 2014) that we currently lack sufficient observations to fully constrain estimates of wetland extent produced via land surface process models and that such differences in modelled wetland extent can account for an uncertainty of between 30 and 40% in the global wetland $CH_4$ emission estimates (Saunois et al., 2016).
Recent work by Tootchi et al. (2019) provides an excellent summary of the current literature and highlights the differences between the multitude of wetland extent datasets currently in use. Furthermore, our previous work (Parker et al., 2018) has shown that important processes such as river overbank inundation can contribute significantly to wetland $CH_4$ emissions but such processes are often lacking in models and, due to their infrequent nature, are not always represented in wetland extent maps.

Given the consensus that there is currently a lack of observations sufficient to fully constrain the $CH_4$ emissions from more complex land surface models, there is a clear opportunity for a simple, data-driven, ensemble-based model, such as WetCHARTs, that is capable of producing estimates of $CH_4$ wetland emissions.

In this study we perform a thorough, global and regional, evaluation of the WetCHARTs $CH_4$ wetland model ensemble using satellite observations of atmospheric $CH_4$ from the GOSAT satellite with the aim of

1. **Identifying whether WetCHARTs is capable of reproducing the observed wetland $CH_4$ seasonal cycle at both global and regional scale**

   2. **Determining whether we can exploit the ensemble of WetCHARTs data to explain the cause of any discrepancies against observations**

   3. **Improving our understanding of which drivers are the most important for constraining wetland $CH_4$ emissions**
**both spatially and temporally**

   4. **Determining whether WetCHARTs can be used as a suitable benchmark against which to assess more complex process-based land surface models**



**4-digit code describes ensemble member - ABCD**

| A | 1 | 2 | 3 | |
|---|---|---|---|---|
| Global Scale Factor (Tg CH$_4$/yr) | 124.5 | 166 | 207.5 | |
| **B** | **1-8** | | **9** | |
| Heterotrophic Respiration Model | MsTMIP Models | | CARDAMOM | |
| **C** | **1** | **2** | **3** | |
| Temperature Dependence | q10 = 1 | q10 = 2 | q10 = 3 | |
| **D** | **1** | **2** | **3** | **4** |
| Extent Parameterisation | SWAMPS & GLWD | SWAMPS & GLOBCOVER | PREC & GLWD | PREC & GLOBCOVER |

**Figure 1.** Description of the 4-digit code used throughout this work to identify the configuration of the WetCHARTs ensemble members. For the extended WetCHARTs period of 2009-2018, there is only one Heterotrophic Respiration Model (CARDAMOM) and two wetland extent parametrisations (using ERA-Interim precipitation) available, giving a total of 18 different configurations.

## 2 The WetCHARTs Ensemble

WetCHARTs (Bloom et al., 2017b) is a wetland CH$_4$ emission dataset derived from satellite-based surface inundation extent and precipitation reanalyses, model-based heterotrophic respiration and a range of temperature dependencies. WetCHARTs has been used in a range of studies (including Parker et al. (2018); Treat et al. (2018); Sheng et al. (2018); Lunt et al. (2019); Maasakkers et al. (2019)) typically as the wetland CH$_4$ a priori in global/regional flux inversion experiments. This study uses v1.2.1 of WetCHARTs which extends the ensemble to 2017 and has improved North American wetlands over previous versions.

Fundamentally, WetCHARTs works by calculating spatially ($x$) and temporally ($t$) resolved CH$_4$ fluxes at a 0.5° × 0.5° resolution globally using the following equation:

$$F(t,x) = sA(t,x)R(t,x)Q_{10}^{\frac{T(t,x)}{10}} \tag{1}$$

where $A(t,x)$ is the wetland extent fraction, itself given by $A(t,x) = w(x)h(t,x)$ with $w(x)$ being the static wetland extent fraction and $h(t,x)$ being the temporal variability. $R(t,x)$ is the heterotrophic carbon respiration per unit area. The term $Q_{10}^{\frac{T(t,x)}{10}}$ represents the temperature dependence of the CH$_4$:C ratio with $Q_{10}$ being the relative CH$_4$:C respiration for a 10°C increase and $T(t,x)$ being the surface skin temperature. $s$ is a global scaling factor. Many other studies and wetland emission models utilise some form of this equation to estimate methane wetland fluxes (Gedney et al., 2004; Clark et al., 2011; Xu et al., 2016; Poulter et al., 2017; Comyn-Platt et al., 2018).

Figure 1 shows the configurations of the WetCHARTs ensemble members used in this study and also provides guidance on the 4-digit identification which will be used to describe the individual ensemble members from hereon.



The global scale factor ($s$ in Equation 1) takes values of 124.5, 166 and 207.5 Tg CH$_4$ yr$^{-1}$ which represent the mean 2000-2009 wetland emission estimates from Saunois et al. (2016) along with a $\pm 25\%$ uncertainty. The full ensemble (FE) utilises nine heterotrophic respiration models for 2010 (Huntzinger et al., 2013) but the extended ensemble (EE) used here (2009-2017) is limited to the CARDAMOM data-constrained terrestrial carbon cycle analysis (Bloom et al., 2016) to calculate

values for $R$. The temperature dependence spans three values for $Q_{10}$, ranging from 1 (i.e. no temperature dependence) to 3 (high temperature dependence). Finally, the wetland extent parametrisation ($A$) for the extended ensemble uses normalised monthly mean ERA-Interim precipitation (Dee et al., 2011) with a static wetland map taken from either the Global Lake and Wetland Database (GLWD - Lehner and Döll (2004)) or GlobCover (Bontemps et al., 2011) to give spatially and temporally varying wetland extent.

In total, for the extended ensemble covering 2009-2017, the above results in 18 ensemble members ($3 \times s$, $1 \times R$, $3 \times Q_{10}$, $2 \times A$). For more extensive details on the ensemble members see Bloom et al. (2017b).

Throughout this study we refer to each ensemble member by a 4-digit code (Figure 1) and highlight these in bold italics in the text. For example, the ensemble member with a global scale factor of 166 Tg CH$_4$ yr$^{-1}$, using CARDAMOM for its heterotrophic respiration, with a temperature dependence of $Q_{10} = 3$ and using precipitation with the Global Lake and Wetland

Database to define the extent would be identified as **2933**. When referring to the parameter but not a specific configuration, the **xxCx** nomenclature is used (e.g. temperature dependency) and when referring to a specific value for a set of configurations the **xx1x** nomenclature is used (e.g. all ensemble members with a $Q_{10} = 1$ temperature dependency).

## 3   TOMCAT Model Simulations

In order to compare the CH$_4$ emissions from WetCHARTs with atmospheric CH$_4$ observations, we process the emissions

through a global 3-D atmospheric chemistry transport model, TOMCAT (Chipperfield, 2006). Throughout this study, when we refer to CH$_4$ concentrations from WetCHARTs, we are referring to the output from TOMCAT simulations using the WetCHARTs wetland emissions. These TOMCAT simulations are performed globally at 1.125° horizontal resolution between 2009 and 2017 using ERA-Interim meteorology to force the model (Dee et al., 2011). The WetCHARTs ensemble is used as the surface wetland emissions. Each ensemble member from WetCHARTs, as described in Section 2, is used to simulate a

separate CH$_4$ tracer, along with a reference CH$_4$ tracer that has all other CH$_4$ emission sectors included apart from wetland emissions, resulting in 19 model tracer simulations. The non-wetland CH$_4$ fluxes are kept consistent between all simulations, using EDGAR (v4.2) for anthropogenic emissions and GFED (v4.1s) for biomass burning. Prescribed annually-repeating values taken from Yan et al. (2009) are used for rice paddy emissions, with the remaining emissions (oceans, termites) used as described in Patra et al. (2011). The atmospheric sink is included via annually-repeating atmospheric OH and O($^1$D) fields and

the methanotrophic soil sink is included as in McNorton et al. (2016).





## 4 GOSAT Proxy XCH$_4$ Data

This study uses satellite observations of total column dry air mole fractions of CH$_4$ (XCH$_4$) generated by the University of Leicester Proxy XCH$_4$ GOSAT retrieval (Parker et al., 2011, 2015, 2020) as part of the ESA Climate Change Initiative (Buchwitz et al., 2017) and the Copernicus Climate Change Service (Buchwitz et al., 2018).

The GOSAT XCH$_4$ data has been extensively validated (Dils et al., 2014; Parker et al., 2015; Buchwitz et al., 2017; Parker et al., 2020), primarily using data from the Total Carbon Column Observing Network (TCCON). TCCON is a global network of ground-based, high resolution Fourier transform spectrometers recording direct solar spectra in the near-infrared spectral region (Wunch et al., 2011). The TCCON data are tied to World Meteorological Organization (WMO) standards (Wunch et al., 2010) and are the primary validation data for satellite observations of greenhouse gases (Cogan et al., 2012; Wunch et al., 2011;
Dils et al., 2014).

This version of the GOSAT Proxy XCH$_4$ data (v7.2) agrees well with TCCON data, with an overall bias of 0.32 ppb, a standard deviation of 13.64 ppb and a correlation coefficient of 0.91 between the 73,304 coincident GOSAT-TCCON measurements. GOSAT XCH$_4$ has additionally been validated against aircraft measurements over the Amazon (Webb et al., 2016).

This data has been heavily utilised by the atmospheric inversion community and many studies (Fraser et al., 2013; Cressot
et al., 2014; Turner et al., 2016; Alexe et al., 2015; Berchet et al., 2015; Feng et al., 2017; Ganesan et al., 2017; Sheng et al., 2018; Maasakkers et al., 2019; Saunois et al., 2020; Lunt et al., 2019) have used this data to successfully infer regional and global emissions of CH$_4$.

## 5 Global Evaluation of WetCHARTs

This section evaluates the WetCHARTs ensemble by computing global statistics comparing the TOMCAT model simulations
against the GOSAT XCH$_4$ observations. In order to properly compare the model to observations, the model simulations are all sampled at the time and location of the GOSAT measurement and a total column model XCH$_4$ value is computed with the GOSAT averaging kernel applied. Globally between 2009-2017 we have 3,382,474 GOSAT XCH$_4$ observations, with 2,075,699 over land and 1,306,775 over ocean. In this study we restrict our analysis to the observations over land to focus on CH$_4$ emissions.

To assess how representative the WetCHARTs emission ensemble is of the observed CH$_4$ wetland seasonal cycle, we calculate the smoothed detrended seasonal cycle by applying the NOAA CurveFit (Thoning et al., 1989) to the XCH$_4$ data for the GOSAT observations and for each ensemble member. We also apply this routine to the TOMCAT model simulation that has no wetland CH$_4$ emissions included. The seasonal cycle from this "no wetland" simulation is subtracted from all other seasonal cycles, resulting in data representing only the wetland component of the seasonal cycle. This makes the assumption
that wetlands dominate the uncertainty in interannual variability of the CH$_4$ emissions and the remaining CH$_4$ sources are in comparison far less uncertain.

We first perform this analysis globally and in later sections separately for data within each region of interest. Figure 2 shows the correlation coefficients between the wetland seasonal cycles when calculated globally. The left column shows the



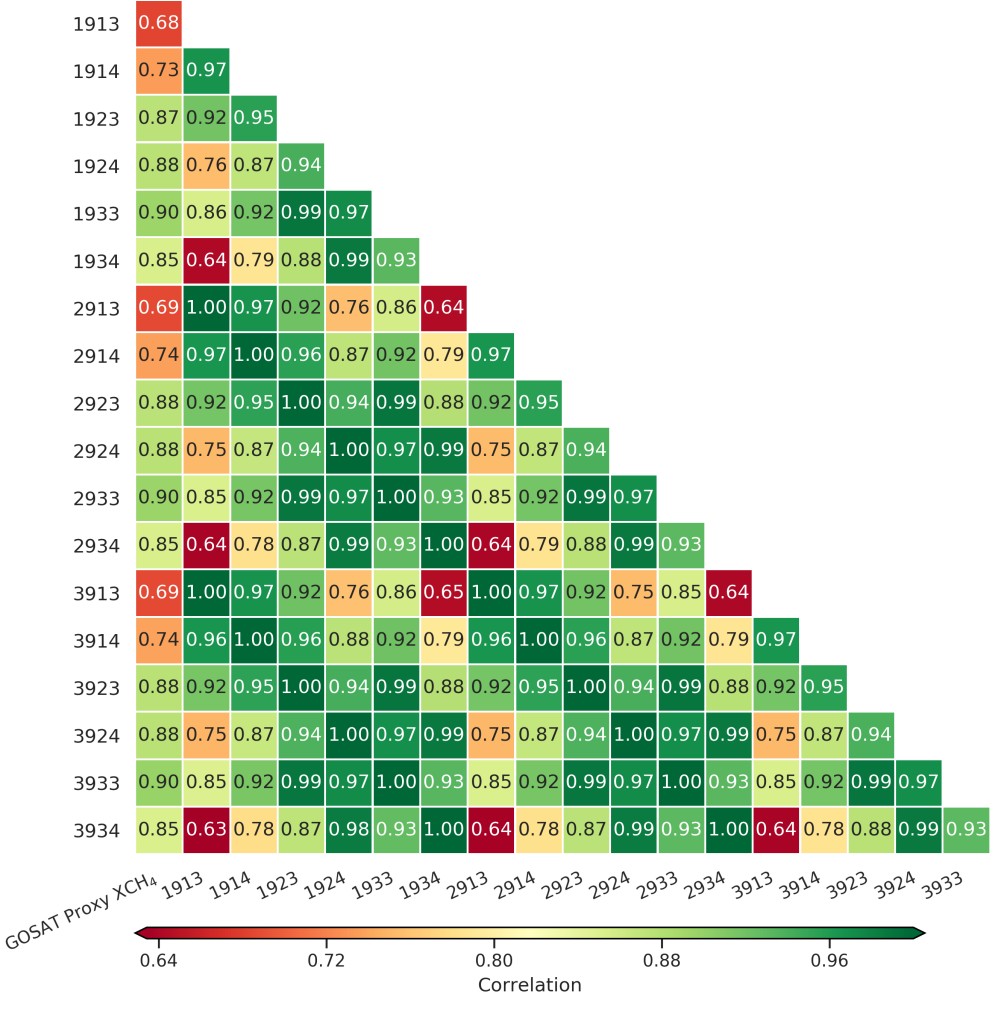

**Figure 2.** Global correlation coefficient between the wetland seasonal cycle derived from the GOSAT Proxy XCH$_4$ and each WetCHARTs ensemble member (left-most column) and also between individual ensemble members.

correlation against the GOSAT-derived seasonal cycle whilst the remaining columns show the model-model correlations for different pairs of ensemble members. The correlation against observations highlights that when considered at a global scale, the temperature dependence is clearly important. The ensemble members where there is no temperature dependence, (i.e. $Q_{10}$ = 1, ensemble members *xx1x*), all correlate far more poorly to observations than the same configuration including temperature

5    dependency (e.g. r = 0.69 for *2913* vs r = 0.88 for *2923*). While it is clear that some temperature dependence is important, it is not clear from analysis at a global scale what degree of temperature dependence gives the best agreement with observations. Instead, regional analyses as performed in Section 6 are required.





The other feature of note in Figure 2 is the variability in the inter-ensemble correlation coefficients. For example, correlations between pairs of WetCHARTs-based TOMCAT simulations can be as low as 0.63 (*3934* vs *1913*). In fact, the correlation coefficient is extremely poor (r = 0.63-0.65) for all simulation pairs where the temperature dependency is at the opposite extreme and the alternate wetland extent dataset is used. This further reinforces the need to continue this analysis at a regional scale to understand the driving factors for these differences over a range of wetland ecosystems.

## 6 Regional Evaluation of WetCHARTs

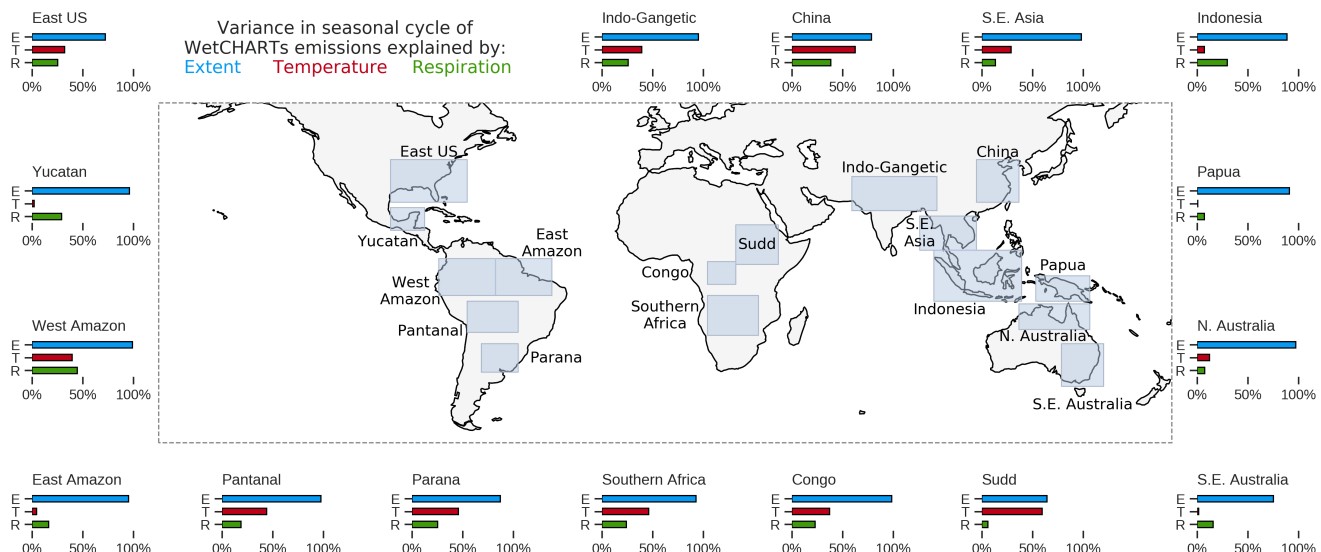

**Figure 3.** Explained variance ($R^2$) calculated as the individual partial correlation between the seasonal cycle of the WetCHARTs CH$_4$ emissions and the seasonal cycles in the wetland extent fraction, temperature dependency and heterotrophic respiration for each region that were used to derive the WetCHARTs emissions. These values are averaged across all ensemble members to give an indicative value for the sensitivity of each region to the different driving parameters. Note that due to potential cross-correlations, it is not necessarily expected that the percentages sum to 100%.

Before performing a detailed regional evaluation, it is useful to consider which of the varying WetCHARTs ensemble parameters is responsible for the variation in the subsequent WetCHARTs emissions. For this purpose we perform a variance analysis by calculating the partial correlation (Vallat, 2018) between timeseries for each parameter and the resulting WetCHARTs emissions. We average the result across all ensemble members to derive an indication of which drivers (extent, temperature and respiration) are important in which regions (Figure 3). We also use this figure as an opportunity to identify the geographic extent of the different wetland regions used throughout this study.

When considered globally, no one parameter is found to be the primary driver of the variation in the WetCHARTs emissions with the R$^2$ value ranging from 29-37%. For the Southern Hemisphere however, the wetland extent fraction itself is found to





explain 83% of the variation in the emissions, with the temperature dependency and respiration both explaining approximately a third on their own.

When performing this analysis in smaller regions, individual behaviours become more apparent. For example, variations of heterotrophic respiration are found to be much more important in some regions such as the West Amazon where they can

5   explain 45% of the observed emissions, whereas for the Sudd region it only explains 6% of the variation in the emissions. Likewise, the temperature dependency explains 63% of the variance in China and over 40% in many regions (West Amazon, Pantana, Paraná, Sudd and Southern Africa) but only a few percent in other regions (Yucatán, East Amazon, Indonesia, Papua, S.E. Australia).

The wetland extent is found to be the dominant explanation for the variance in all regions, often by a large margin, explaining

10   > 95% of the variance in the Yucatán, West Amazon, East Amazon, Pantanal, Congo, Indo-Gangetic, S.E. Asia and N. Australia regions. Even for the regions where the wetland extent explains the least variance such as East US (72%), Sudd (64%), China (79%) and S.E. Australia (75%), it still explains more than the other parameters.

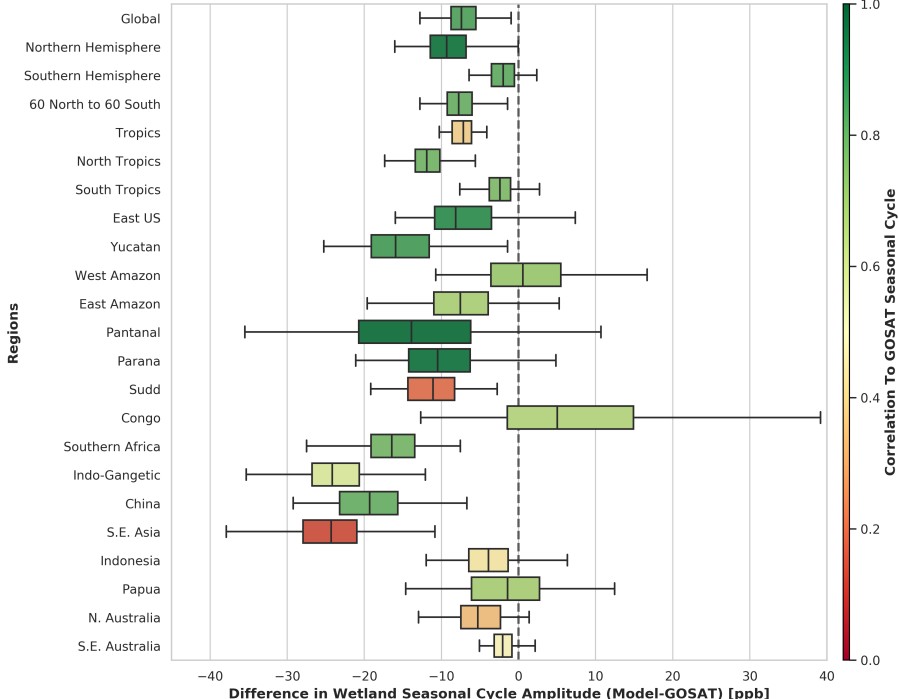

**Figure 4.** Distributions of the wetland seasonal cycle amplitude difference between the modelled WetCHARTs and GOSAT observations for all ensemble members and all years for the different regions. The data is coloured by the average value of the correlation coefficient between the modelled and observed wetland seasonal cycle.

The capability of the WetCHARTs emissions to successfully represent the observed wetland $CH_4$ seasonal cycle amplitude is a vital component in the assessment of the utility of the emissions. In addition to assessing the seasonal amplitude we assess



the magnitude and phase of the emissions using the correlation coefficient between the simulated and observed seasonal cycles. Figure 4 shows the distributions of the wetland seasonal cycle amplitude difference between model and observation for all ensemble members and all years for the different regions. Furthermore, each bar is coloured by the ensemble-average correlation coefficient between the wetland seasonal cycle of the WetCHARTs ensemble members and the GOSAT observations.

There is a clear hemispheric distinction between the difference in seasonal cycle amplitude, with the underestimation being far more pronounced in the Northern Hemisphere (9.3 ppb) than in the Southern Hemisphere (1.9 ppb). This is further emphasised by considering the tropical region only, the North Tropics underestimates the seasonal cycle amplitude by 11.9 ppb, compared to just 2.4 ppb in the South Tropics.

When considering comparisons on a regional scale, it is possible to characterise the different regions into groups that exhibit
similar behaviour.

For some regions the seasonal cycle amplitude is always significantly underestimated for all years and for all ensemble members. This is particularly the case for Southern Africa, China, S.E. Asia and the Indo-Gangetic region where WetCHARTS underestimate the observed seasonal cycle by median values of -16.6, -19.1, -23.1 and -24.2 ppb respectively.

This however is not the case for all regions with some regions such as West Amazon, Papua, Indonesia and the Congo
exhibiting a small median difference in the seasonal cycle (+0.7, -1.2, -4.0 and +5.0 ppb respectively) but with a large variability between years and ensemble members. The Congo region in particular exhibits a large variability in the seasonal cycle amplitude difference (-12.7 - 52.4 ppb) despite the reasonable correlation to the observed seasonal cycle (r = 0.67).

Some regions (East US, Pantanal, Paraná, Yucatán) have a high correlation to the observed seasonal cycle (0.88, 0.94, 0.93 and 0.84) but consistently underestimate the amplitude (-7.7, -14.1, -10.5 and -15.8 ppb) and also contain a large amount of
variability between the different ensemble members and years.

Finally, the underestimation of the seasonal cycle amplitude can be similar to those above (-11.2 ppb) but with a very poor correlation to observations (r = 0.20), as observed for the Sudd region.

In order to understand the effect that the different WetCHARTs parametrisations have on the different regional emissions, the correlation coefficient and standard deviation between the simulated and observed seasonal cycle is calculated for each region
for each ensemble member. The full table of correlation coefficients per region per ensemble member is presented in Appendix A (Figure A1). To highlight and isolate the individual effects of adjusting the 3 driving parameters (global scale factor, temperature dependency and wetland extent map), we plot the correlation coefficient for each configuration of the ensemble and link together data points where the only change between the ensemble members is the change in the specified parameter. This is demonstrated for the Tropics region in Figure 5. In this figure we see that the correlation coefficient is largely unaffected by
a change in the global scale factor, when controlling for the other parameters. In contrast, a temperature dependency of $Q_{10}$ = 2 clearly leads to a higher correlation coefficient for the Tropics, when controlling for the other parameters, with $Q_{10}$ = 3 leading to the worst correlation coefficient in all cases. Similarly, it is evident that the GLWD wetland extent map, performs significantly better than the GlobCover map, significantly increasing the correlation coefficient in all cases.

To emphasise the effect related to the relative change of each parameter when considering the behaviour across multiple
regions, we subtract as a baseline the lowest correlation coefficient from each set of lines. This change in correlation coefficient



**Figure 5.** For each of the three parameters that vary within the ensemble (global scale factor, temperature dependency and wetland extent map), we plot the correlation coefficient between the modelled and observed wetland seasonal cycle. We join data points where the other two parameters are kept constant and the only change is due to the specified parameter. Colours indicate the different groupings. For example, for temperature dependency, we join the 3 data points where the temperature dependency varies between $Q_{10} = 1$, $Q_{10} = 2$, $Q_{10} = 3$ but the wetland map and the global scale factor are the same for the 3 joined data points. This allows the influence of the change in each individual parameter to be assessed.

therefore gives an indicator of the improvement obtained by the change in each parameter, while keeping the rest of the configuration for that ensemble member the same. Figure 6 shows the distribution of this change in correlation coefficient across all regions. These results illustrate that the choice of global scale factor makes little difference to the correlation between the observed and modelled wetland seasonal cycle. In contrast, the choice of $Q_{10}$ value is found to make a substantial difference

5   to the correlation coefficient. Setting $Q_{10} = 1$ typically produces the worst correlation coefficient (as indicated by a median value of 0 improvement), with $Q_{10} = 2$ typically leading to the largest improvement in correlation coefficient (with a median value of 0.067 and up to an improvement of 0.20). On average, $Q_{10} = 3$ does not perform as well in a general sense (a median value of 0.017) but it does produce a large spread in values indicating that for some regions it does lead to better performance. Finally for the wetland extent, it is very clear that the GLWD wetland extent map performs better for the majority of regions,

10   leading to an improvement (75th percentile) of 0.069 in the correlation coefficient compared to to just 0.015 for GlobCover.

As well as the correlation between the modelled and observed seasonal cycle, the standard deviation between the two is a useful metric for determining how well the WetCHARTs emissions allow the seasonal cycle to be reproduced. In Figure 7 we



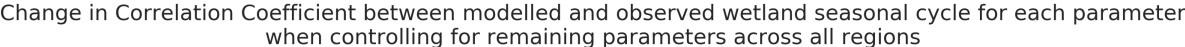

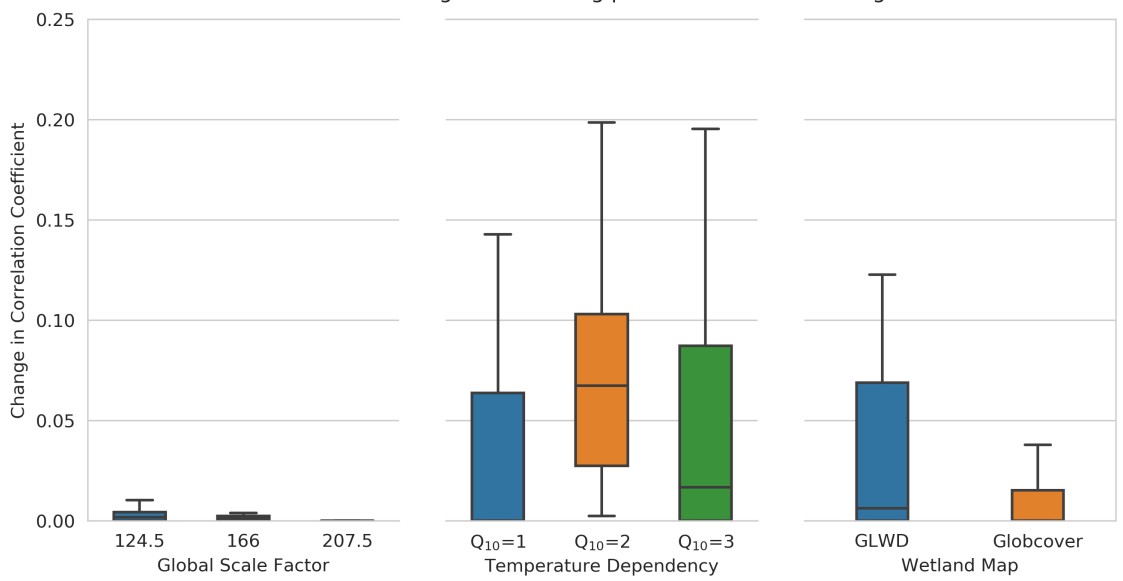

**Figure 6.** The change in correlation coefficient between the modelled and observed wetland seasonal cycle related to each parameter with all other parameters kept constant. The change is calculated as the improvement in correlation coefficient from the lowest value from each set of joined ensemble members. The distribution of this change in correlation coefficient across all regions is illustrated using a box-and-whisker plot with the quartiles indicated by the boxed area and whiskers indicating the remaining data (excluding extreme outliers).

examine the increase in standard deviation related to each parameter, with a smaller value indicating better performance. In contrast to the analysis of the correlation, we find that the choice of global scale factor does have an impact here. The global scale factor of 124.5 Tg yr$^{-1}$ has the lowest median change in standard deviation (0.0 ppb, indicating it is always the lowest/best on average) but exhibits a large spread across regions (75th percentile of 0.82 ppb). The medium global scale factor of 166 Tg yr$^{-1}$ is much more consistent across regions with a smaller spread (75th-25th range of 0.42 ppb vs 0.82 ppb) but on average produces a larger standard deviation than the lower scale factor (median value of 0.31 ppb). For temperature dependency, the picture is clearer, with a $Q_{10} = 2$ producing the smallest median standard deviation (0.016 ppb) and the smallest spread (75th-25th range of 0.13 ppb). Finally, the GLWD wetland map is found to perform emphatically better than the GlobCover map for nearly every region, with GlobCover on average worsening the standard deviation by an average 0.34 ppb and up to over 4 ppb in some cases

When considering the two metrics (correlation coefficient and standard deviation) in unison, a consistent conclusion can be drawn that a temperature dependency can cause large changes in the agreement between model and observation, with a $Q_{10} = 2$ value typically performing better on average but $Q_{10} = 3$ performing better for some regions. Both results suggest that some temperature dependency is necessary, i.e. that $Q_{10} = 1$ performs worse than the alternatives. Likewise, the GLWD wetland extent map is consistently found to perform better than the GlobCover map. Whilst the global scale factor is found to have little





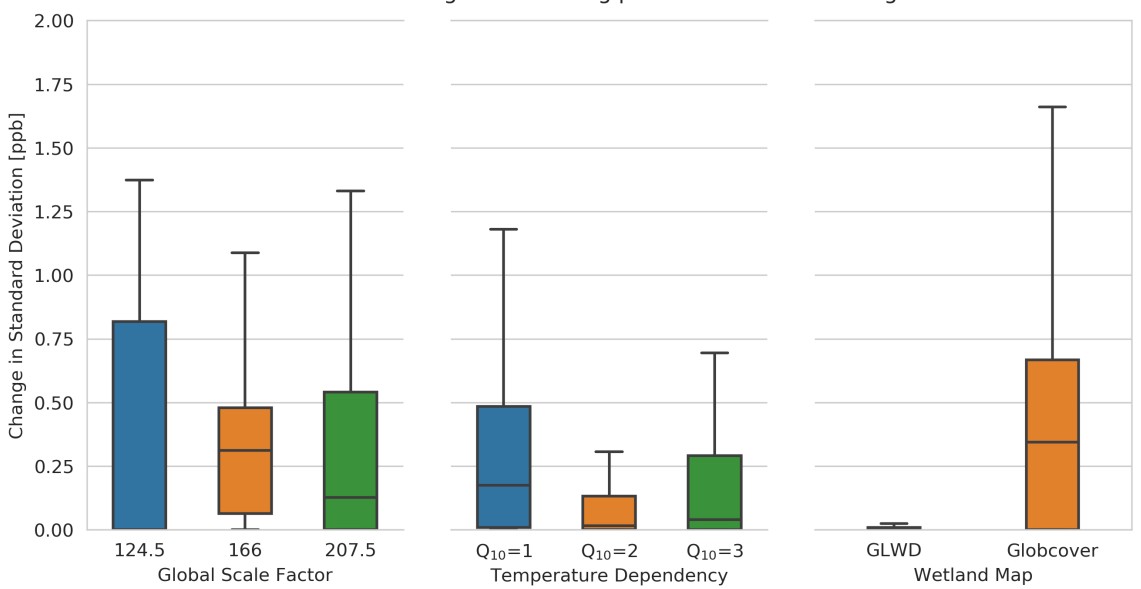

**Figure 7.** The change in standard deviation (ppb) between the modelled and observed wetland seasonal cycle related to each parameter with all other parameters kept constant. The change is calculated as the difference in standard deviation from the lowest value from each set of joined ensemble members. The distribution of this change in standard deviation across all regions is illustrated using a box-and-whisker plot with the quartiles indicated by the boxed area and whiskers indicating the remaining data (excluding extreme outliers).

influence on the correlation coefficient, it does influence the standard deviation between the modelled and observed wetland seasonal cycle.

This summary of the regional analysis all points towards complex interactions and highlights the difficulty in using a simple parameterisation to represent many complex inter-related processes. A more detailed analysis of exemplar regions as case studies is valuable in explaining the above behaviours in more detail.

## 7   Case Study 1: The Paraná River

In Parker et al. (2018) we identified the Paraná River as a particularly interesting example of a case where excess precipitation led to severe overbank inundation, which was clearly evident in multiple ancillary datasets (MODIS visible imagery, GRACE Terrestrial Water Storage Anomaly, etc). This large but temporary increase in wetland extent produced significant methane emissions which were clearly observed from satellite observations but which were missing from any model simulations as this inundation mechanism was not represented in the underlying emission datasets.

Our previous analysis identified a large anomalous event in 2010 but the analysis ended at 2015. For this work we are able to extend this analysis until the end of 2017. The monthly mean difference between the simulated and observed wetland seasonal



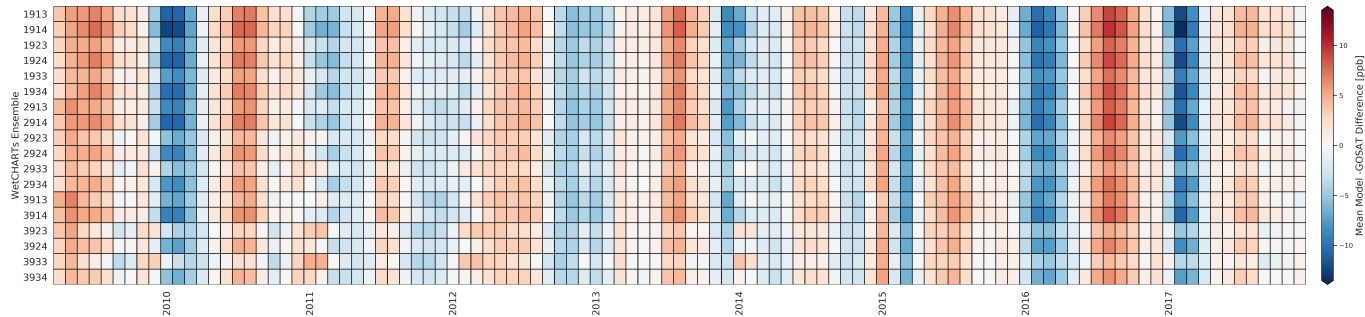

**Figure 8.** Modelled - GOSAT monthly mean wetland $CH_4$ seasonal cycle differences for all WetCHARTs ensemble members from 2009 to 2017 over the Paraná region.

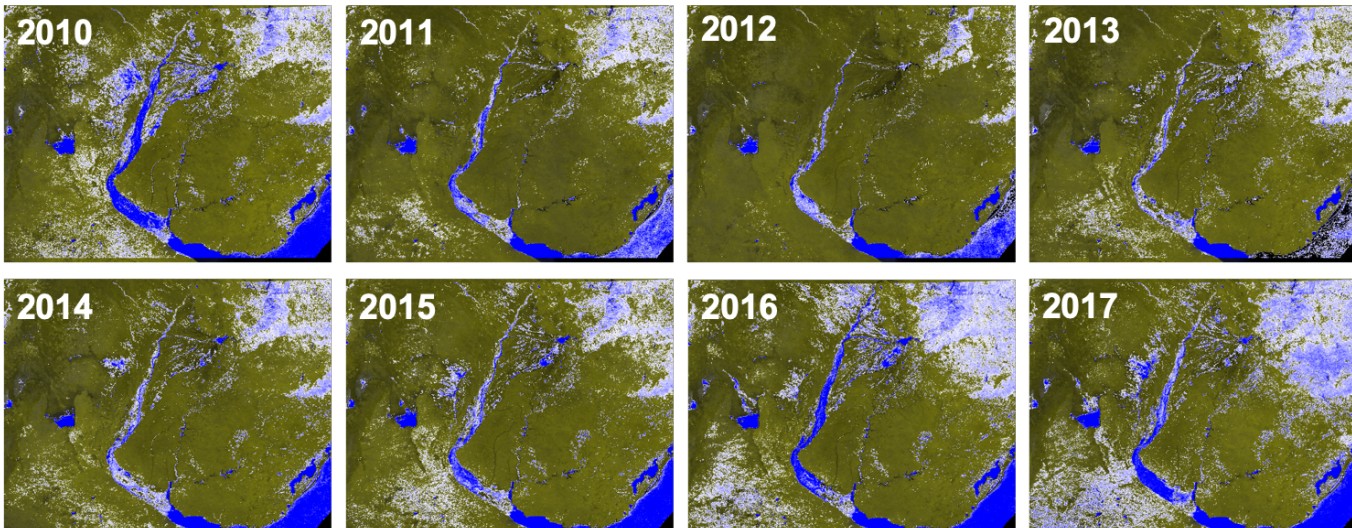

**Figure 9.** MODIS Normalized Difference Water Index (NDWI) indicating surface water extent for November-April from 2010 to 2017 overlaid onto MODIS RGB imagery for the Paraná region.

cycles over this region are shown in Figure 8. The large 2010 anomaly, where simulations are all significantly less than the observation, is again observed but in addition, we observe similarly large anomalies in 2016 and 2017. These large anomalies are persistent across all model ensembles but are at a minimum for the more intense emission scenarios (large scale factor and larger temperature dependency) and when the GLWD wetland extent parameterisation is used.

5    Comparison to MODIS NDWI (Fig. 9) explains the nature of the anomalies observed above. For 2010, 2016 and 2017 there were significant increases in river inundation along the channel of the Paraná River, especially in the Paraná Delta region to the north of Buenos Aires. The resulting $CH_4$ emissions from this increase in wetland extent are clearly observed by GOSAT but are not represented within the WetCHARTs model ensemble, leading to the model ensemble monthly mean under-





representing the observed CH$_4$ amount by up to -8.5, -8.2 and -9.4 ppb for 2010, 2016 and 2017 respectively. For the remaining years of the analysis (2011-2015) which correspond to the years where the NDWI extent is noticeably lower, the agreement between the WetCHARTs ensemble and the observation data is far better with maximum differences of -3.5, -4.7, -5.8, 3.0 and -7.5 ppb respectively. Furthermore the annual range of standard deviations of the ensemble monthly mean differences for

5  2010/2016/2017 is considerably higher (3.9 - 5.4 ppb) than for the 2011-2015 period (2.3 - 3.7 ppb). This again highlights the increased variability during 2010, 2016 and 2017 and the characterisation of these years as anomalous, driven by the significant change in observed wetland extent related to the inundation of the Paraná River and Paraná Delta.

**This case highlights one limitation of WetCHARTs, namely that there is no underlying hydrological model to account for river flow or inundation but instead local precipitation determines the wetland extent variability. As such,**

10  **WetCHARTs may not capture the behaviour where wetland extent is determined by behaviour upstream. This however allows WetCHARTs to act as a baseline against which to assess such behaviour in land surface models and determine if they are out-performing the simpler WetCHARTs precipitation-driven approach.**

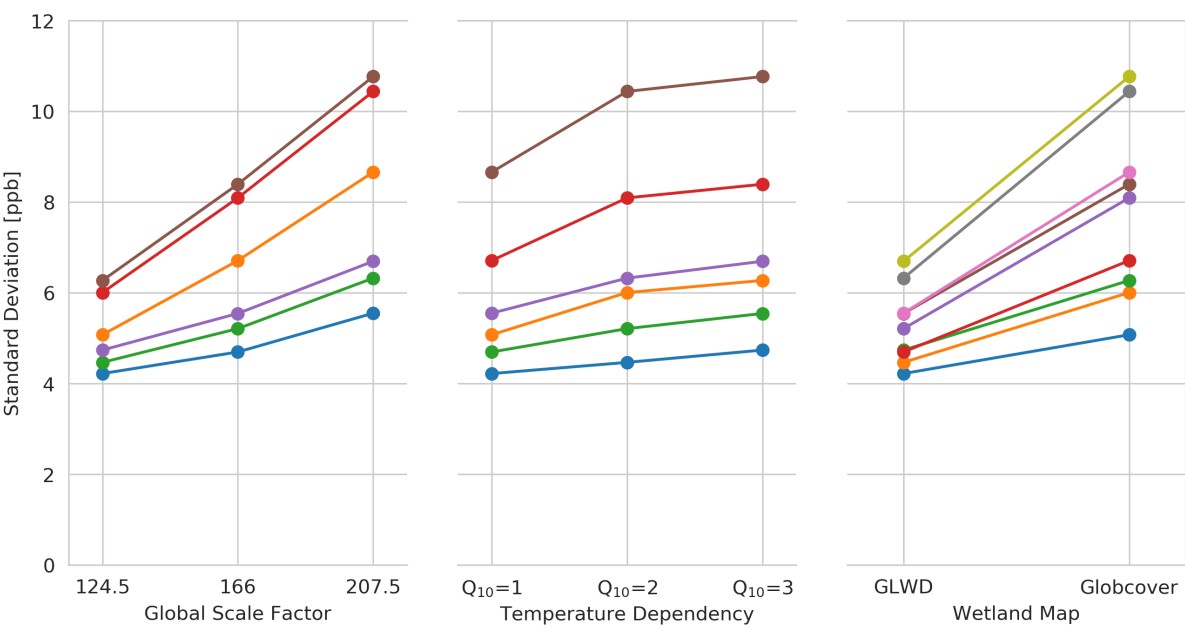

**Figure 10.** The standard deviation between the modelled and observed wetland seasonal cycle ror each of the three parameters that vary within the ensemble (global scale factor, temperature dependency and wetland map). Data points are joined together where the other two parameters are kept constant and the only change is due to the specified parameter. This allows the influence of the change in each individual parameter to be assessed.





## 8 Case Study 2: The Congo

The Congo is perhaps one of the most important wetland regions to be able to well-characterise as it has the potential to dominate African methane wetland emissions but is still relatively poorly understood (Lee et al., 2011; Melton et al., 2013; Zhang et al., 2017a; Becker et al., 2018; Lunt et al., 2019).

Figure 4 highlights the huge discrepancy between the observed $CH_4$ wetland seasonal cycle amplitude from GOSAT and that simulated by any of the WetCHARTs ensemble members (a median difference of 5 ppb but ranging from -12.7 to 52.4 ppb). This is consistent with recent work in both Lunt et al. (2019) and Maasakkers et al. (2019) which use WetCHARTs data as the prior in atmospheric flux inversions. In Lunt et al. (2019), flux inversions of atmospheric $CH_4$ over Africa suggest that the mean WetCHARTs emissions over the Congo region are far too high (8.5 Tg yr$^{-1}$ in total with 90% of this attributed to
wetlands) and need to be reduced significantly to be consistent with observations in the range of 2.7-4.1 Tg yr$^{-1}$. This is also more consistent with Tathy et al. (1992) who estimate methane emissions within the flooded Congo Basin using flux chamber measurements of 1.6-3.2 Tg yr$^{-1}$. Maasakkers et al. (2019) (Figure 4 therein) shows similarly that the posterior emissions after inversion of the GOSAT data require a reduction to the original WetCHARTs prior emissions over the Congo region.

Figure 10 demonstrates that the standard deviation of the Model-GOSAT difference is substantially worsened when either
the temperature dependency (***xxCx***, increasing from a median of 5.31 ppb to 6.48 ppb) or the global scale factor (***Axxx***, increasing from a median of 4.91 ppb to 7.68 ppb) are increased. This is indicative of there already being too much $CH_4$ from WetCHARTs in this region and any parameterisation that enhances it further (either by scaling or adding a temperature dependency) exacerbates the discrepancy. This points to the wetland extent fraction being too large and this wetland extent masking clearly plays a significant role as the standard deviation differs substantially based on which extent mask is being
used; ranging from 4.22 - 6.70 ppb for the GLWD-based ensemble members but 5.08 - 10.77 ppb for the GlobCover-based simulations. The different wetland extent masks however do not affect the correlation coefficient between the simulated and observed seasonal cycles (Fig A1), suggesting that both wetland masks are as capable in parameterising the observed seasonal cycle but differ in the magnitude of the resulting emissions. Figure 12 compares both wetland extent datasets against the JRC Surface Water Occurrence and Maximum Extent datasets (Pekel et al., 2016), confirming that the wetland extent used in
WetCHARTs is significantly higher than suggested by the JRC data.

Although the spatial resolution of GOSAT is relatively coarse (∼250 km), Figure 11 shows that it is possible to identify the spatial signature of the wetland signal in both the GOSAT observations and the model simulations (sampled at the GOSAT sounding locations). The GOSAT wetland signal (i.e. the difference to the simulation without any wetland emissions) is relatively weak, with a maximum (95th percentile) value of 11.9 ppb. We compare this to WetCHARTs ensemble members ***2923***
and ***2924*** (chosen to be illustrative of the wider ensemble with the medium global scaling factor and medium temperature dependency). The maximum wetland signal from the WetCHARTs ***2923*** and ***2924*** ensemble members is much stronger than that derived from GOSAT, with values of 20.5 and 23.6 ppb respectively. As well as a much smaller maximum signal, the spatial standard deviation of the wetland signal for GOSAT is found to be 6.8 ppb, much smaller than the standard deviations from WetCHARTs of 11.3 ppb (***2923***) and 15.1 ppb (***2924***).





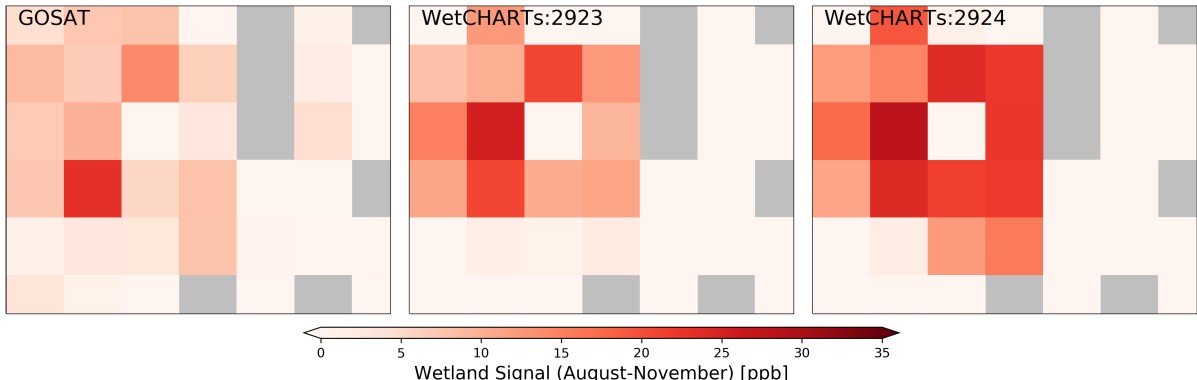

**Figure 11.** The average wetland signal for August-November between 2009-2017 over the Congo region. The wetland signal is defined as the difference between the detrended GOSAT or WetCHARTs model simulations and the simulation with no wetland emissions. Two WetCHARTs ensemble members are shown (*2923* and *2924*) for illustration but all ensemble members exhibit similar behaviour.

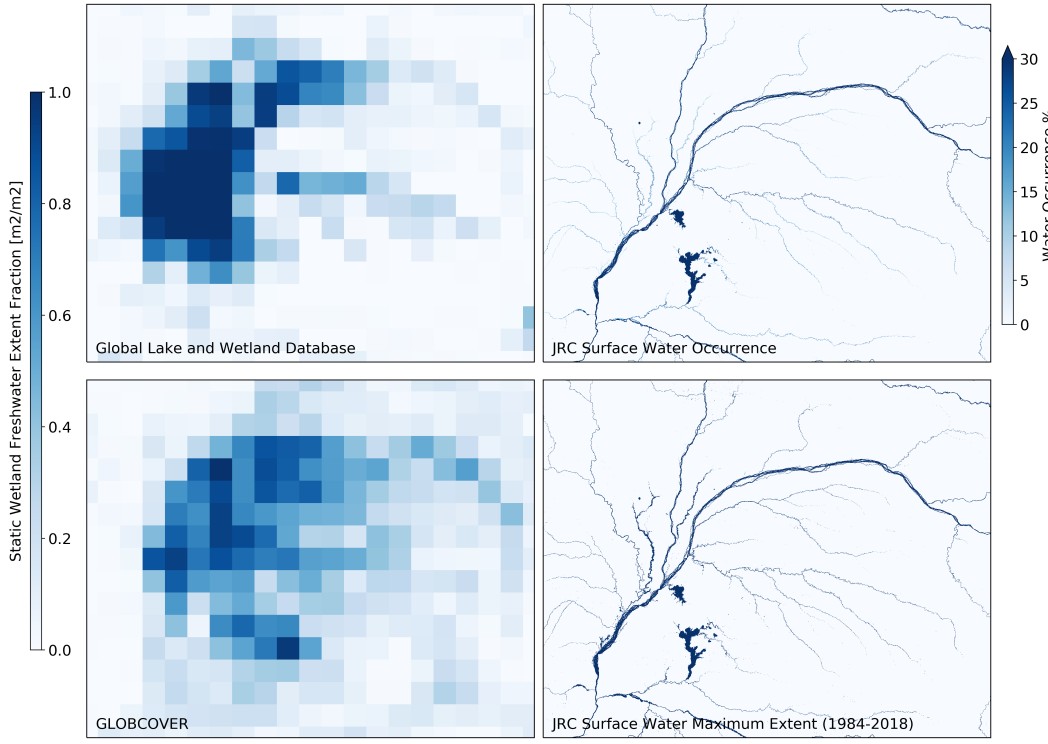

**Figure 12.** Maps over the Congo region showing the GLWD and GlobCover wetland data used to drive WetCHARTs compared to the JRC Surface Water Occurrence and Maximum Extent datasets



This demonstrates that while the spatial signature of the Congo wetland emissions generated by WetCHARTs are reasonably consistent with that from observations, the magnitude and variability of the emissions are far higher than those we observe from GOSAT. This remains the case for the smallest global scale factor and for no temperature dependency, leaving only the wetland fraction or heterotrophic carbon respiration per unit area as tuning parameters to reduce the emissions closer to observations.

**This case highlights that future WetCHARTs development would benefit from further exploration of the characterisation and sensitivity to the heterotrophic respiration, with the extended ensemble currently only featuring a single member (CARDAMOM).**

## 9   Case Study 3: Sudd

In this section we examine the Sudd wetland region. The Sudd wetlands, located in South Sudan, are one of the largest tropical

wetlands in the world and are the largest wetland ecosystem in the Nile basin. They are fed via the White Nile, originating at Lake Victoria to the South with flow through the region ultimately leading in to the Nile to the North. These wetlands therefore play a major role in regional hydrology and understanding their behaviour is of vital importance for environmental, economical and humanitarian reasons.

    The extent of these wetlands is driven by seasonal inundation and outflow from Lake Victoria (Rebelo et al., 2012), albeit

significantly affected by the complexities of the regional hydrology. The wetland extent exhibits a maximum each year between August and November, coincident with the rainy season. This seasonal flooding has been estimated by Rebelo et al. (2012) to increase the wetland extent in the region by a factor of 4, with the total wetland area split between permanent (18%) and seasonally inundated (82%) wetlands. MODIS NDWI data (not shown) shows that as well as the increase in surface water directly over the Sudd wetland region, there is also increased surface water evident further to the north of the region around

Lake Tana and the Blue Nile Basin.

    The results for this region are of particular interest as whilst the discrepancy between the magnitude of the simulated and observed seasonal cycles is comparable to other regions (-11.2 ppb) the correlation coefficient is extremely poor at just 0.2. This suggests that unlike many of the other regions where it is the magnitude of the seasonal cycle that WetCHARTs does not fully represent, this is one of the few regions where the seasonality is also poorly represented. This is illustrated by Figure 13

which shows the GOSAT seasonal cycle over the Sudd region (black) along with the range of the WetCHARTs ensemble (red) for the full seasonal cycle (top panel) and with the "no wetland" simulation subtracted, resulting in a wetland seasonal cycle (2nd panel). This wetland seasonal cycle shows that while observations have a clear seasonal cycle, with the signal ranging from -10.8 to 14.3 ppb, the typical seasonal cycle for the WetCHARTs ensemble is less than half of this (ranging between -6.7 to 6.3 ppb) and importantly does not seem to have any temporal agreement with the observations, resulting in the very poor

correlation coefficients we obtain for all ensemble members.

    The reason for this lack of sufficient seasonality in the WetCHARTs ensemble is evident when examining the time series of underlying driving data for the emissions (Figure 13, lower panels). The seasonality of both the temperature and heterotrophic respiration are out of phase with the wetland extent. This results in either there being sufficient temperature and respiration



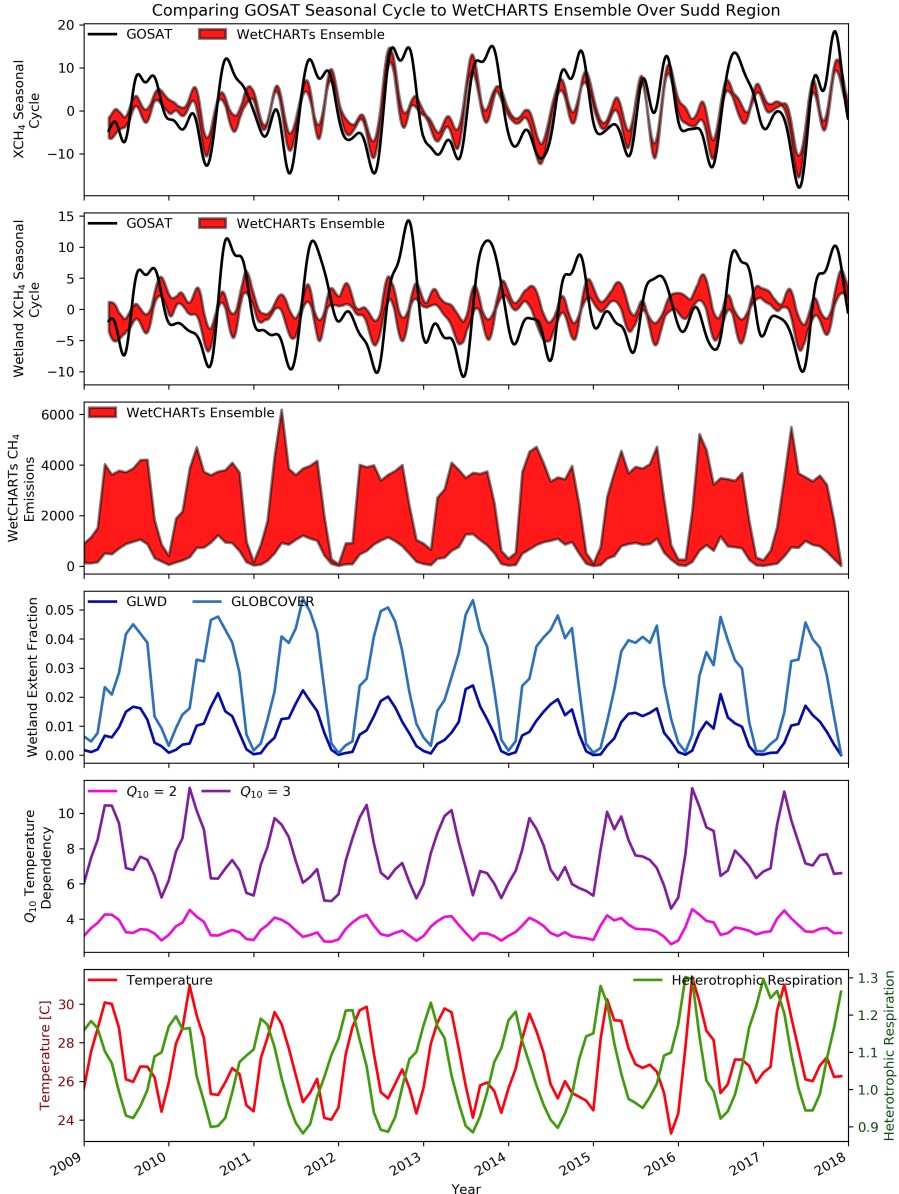

**Figure 13.** Time series over the Sudd wetland region showing the a) the GOSAT and WetCHARTs CH$_4$ ensemble seasonal cycle, b) the wetland component of the above seasonal cycles once the modelled data with no wetland emissions has been subtracted, c) the ensemble of WetCHARTs CH$_4$ emissions, d) the wetland extent fractions, e) the Q$_{10}$ temperature dependencies, f) the temperature and heterotrophic respiration.

to produce methane but no wetland area from which to produce it, or alternatively, a large wetland area but insufficient temperature/respiration to produce the correct magnitude of emissions. The effect of this is very limited seasonality in the CH$_4$



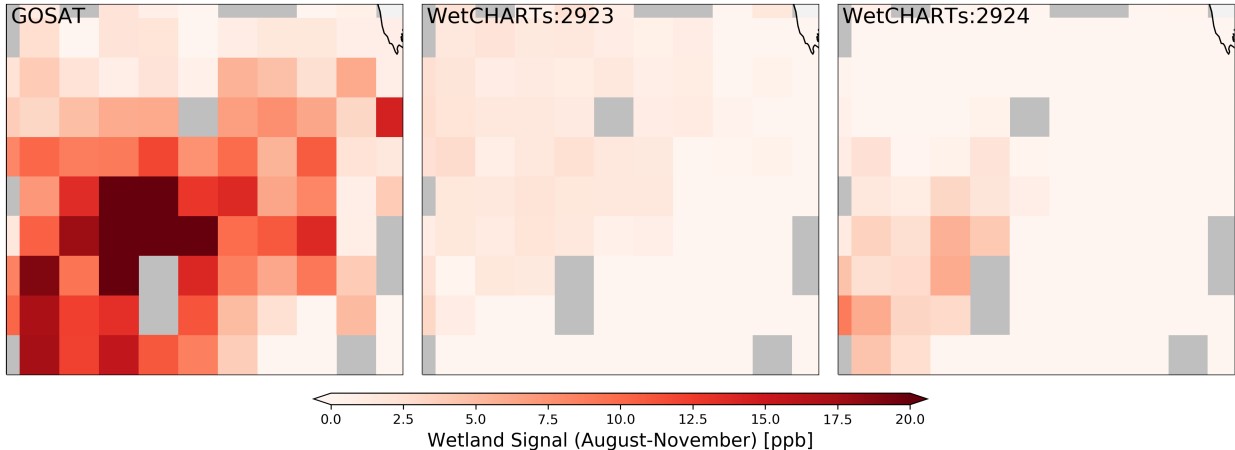

**Figure 14.** The average wetland signal for August-November between 2009-2017 over the Sudd region. The wetland signal is defined as the difference between the detrended GOSAT or WetCHARTs model simulations and the simulation with no wetland emissions. Two WetCHARTs ensemble members are shown (*2923* and *2924*) for illustration but all ensemble members exhibit similar behaviour.

emissions throughout the entire timeseries. This is exemplified in Figure 14 which shows the average wetland signal for August-November (the time period where the satellite $CH_4$ wetland signal peaks) between 2009-2017 for the GOSAT data (left) and two WetCHARTs ensemble members, *2923* (centre) and *2924* (right). Despite the very strong wetland signal observed over this area, directly over the Sudd wetlands, the WetCHARTs signal is extremely low. From Figure 13, the seasonality of the two

wetland extent parametrisations is in agreement with the seasonality of the observed signal, identifying that the issue in this area is not the dynamics of the wetland itself, but rather the seasonality of the temperature and respiration which result in the magnitude of the emissions being far too small even though both wetland extent fraction databases allow WetCHARTs to form wetlands in this area.

    **This case highlights an example where although the wetland extent is sufficient to lead to the correct seasonality in**
**$CH_4$ emissions, the temperature/respiration are out of phase and as such, WetCHARTs can not reproduce the observed variability. This emphasises the importance of the interplay between the different driving parameters and the large discrepancy that can be caused if these are not consistent or sufficiently localised.**

## 10   Case Study 4: Yucatán

The Yucatán area of Mexico contains a variety of wetland ecosystems including mangroves, swamps, marshes and forests with
the watershed containing the Grijalva and Usumacinta rivers in the Tabasco/Campeche region the largest wetland complex in the country. The Pantanos de Centla region, located in the Usumacinta/Grijalva delta, is classified as tropical moist forest and includes permanent wetlands as well as seasonally inundated swamp forests. Mangroves are present between the Pantanos de Centla and the Laguna de Términos to the north, with moist tropical forests to the south, east and west.


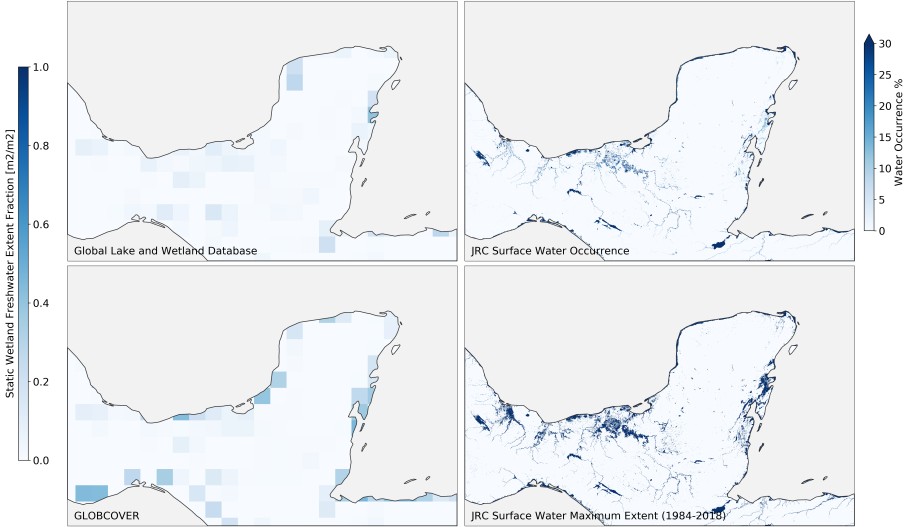

**Figure 15.** Maps over the Yucatán region showing the GLWD and GlobCover wetland data used to drive WetCHARTs compared to the JRC Surface Water Occurrence and Maximum Extent datasets. Neither of the datasets driving WetCHARTs represent the large Tabasco/Campeche wetland complexes in the centre of this region.

For the Yucatán region, the correlation between the observed and simulated wetland seasonal cycles is reasonable, with WetCHARTs ensemble members ranging from 0.76-0.89 (Fig. A1) suggesting that WetCHARTs is capable of representing the phase of the wetland seasonal cycle. However, Figure 4 shows that the wetland seasonal cycle amplitude is consistently underestimated compared to observations, with a median difference of -15.8 ppb and 25/75 percentile values of -19.1 and

5  -11.5 ppb respectively. This underestimation of the seasonal cycle can be attributed to the very low wetland extent fraction from both the GLWD and GlobCover datasets (Figure 15) which fails to represent the wetlands in this region, particularly the large Tabasco/Campeche wetland complexes in the centre of this region, the Alvarado Lagoon system to the west and the Sian Ka'an coastal wetlands to the east. These are barely included in either wetland extent dataset but are clearly identified as being significant from the JRC Surface Water Extent (Figure 15).

10  This case is of particular interest as it is one of the few examples where the Global Lake and Wetland Database performs particularly poorly, not featuring significant wetland extent that relates to a strongly observed wetland signal. The large difference between the wetland extent fraction from GLWD vs GlobCover is also striking (Figure 16) for this region, more so than in the other regions examined. Furthermore, both wetland extent datasets suggest a double-peak in the maximum extent, leading to two peaks in the emission data. GOSAT observations however only typically observes the second of these peaks in most years.

15  **This case highlights the importance of the wetland extent data and that while for the majority of regions it is the detail of the variability in extent that we are concerned over, for some regions the extent is even more poorly constrained with large wetland regions still not being represented.**

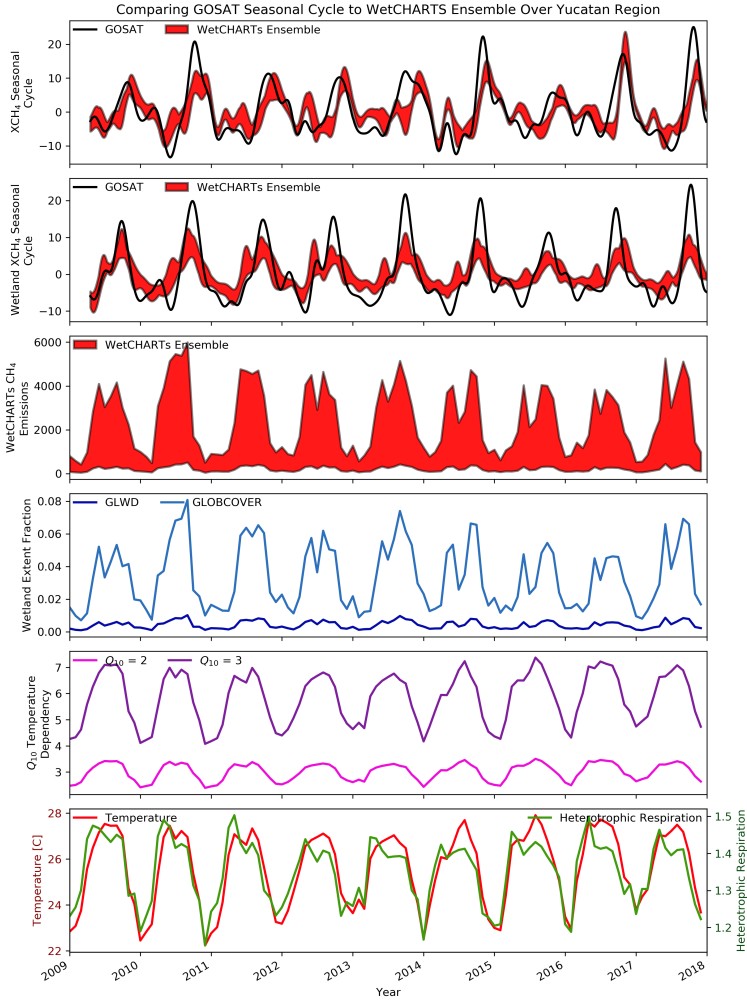

**Figure 16.** Time series over the Yucatán wetland region showing the a) the GOSAT and WetCHARTs CH$_4$ ensemble seasonal cycle, b) the wetland component of the above seasonal cycles once the modelled data with no wetland emissions has been subtracted, c) the ensemble of WetCHARTs CH$_4$ emissions, d) the wetland extent fractions, e) the Q$_{10}$ temperature dependencies, f) the temperature and heterotrophic respiration.

## 11 Discussion and Conclusions

In this study we have assessed the ensemble of WetCHARTs global wetland CH$_4$ emissions against satellite observations. In particular, we have evaluated how well the magnitude and phase of the seasonal cycle of atmospheric CH$_4$ driven by the individual WetCHARTs ensemble members agrees with the seasonal cycle of CH$_4$ observed from the GOSAT satellite.

5    Figure 4 provides an overall summary of how well the phase and magnitude of the observed wetland seasonal cycle can be reproduced by WetCHARTs, both globally and at a regional scale. Across all years and all ensemble members, the observed



global seasonal cycle amplitude is typically underestimated by WetCHARTs by -7.4 ppb but the correlation coefficient of 0.83 shows that the seasonality is well-produced at a global scale. The Southern Hemisphere has less of a bias (-1.9 ppb) than the Northern Hemisphere (-9.3 ppb) and our findings show that it is typically the North Tropics where this bias is worst (-11.9 ppb).

When examining such large geographic areas, there is the possibility that significant positive and negative regional biases cancel each other out. While we find that the majority of individual wetland regions underestimate the seasonal cycle by some degree, we find compensatory effects over central Africa where an underestimation in the seasonal cycle amplitude over the Sudd wetlands (-11.28 ppb) is partially compensated for by an overestimation in the Congo (4.97 ppb). Such an effect has implications for flux inversions over central Africa and we advise caution when interpreting such results.

In our global evaluation, the most significant finding was that the correlation between the modelled and observed $CH_4$ seasonal cycle was substantially higher when the WetCHARTs ensemble includes a temperature-dependency (i.e. when the $Q_{10}$ value is not 1). For equivalent ensemble members (e.g. $Q_{10} = 1$ vs $Q_{10} = 2$ vs $Q_{10} = 3$ with all other parameters the same), the correlation coefficient increased for example from 0.68 (*1913*) to 0.87 (*1923*) to 0.90 (*1933*). As expected, this behaviour at a global scale is not necessarily reproduced for all individual wetland regions with Figure 3 showing that for some regions,

the temperature dependence is far more of a factor in driving the variation in the seasonal cycle than for other regions.

Globally we also find that the difference in the correlation to observations is less reliant on which wetland fraction dataset is used (GLWD vs GlobCover) with both performing reasonably well in representing the global seasonal cycle (correlation coefficients of 0.88 for both *2923* and *2924*). However, the choice of wetland fraction can dominate at regional scales with very significant differences in the correlation coefficient between paired ensemble members (e.g. r = 0.76 for *2923* vs r = 0.46 for

*2924* for the Indo-Gangetic region).

These results all indicate that WetCHARTs is capable of sufficiently reproducing the phase and magnitude of the wetland seasonal cycle in the wider sense, highlighting its utility as an apriori constraint on atmospheric flux inversions (i.e. the use for which it was developed). These results do indicate however that for certain regions, specific ensemble members do perform significantly better than others, whether due to the temperature dependence or wetland extent parametrisation. This therefore

highlights that for focused regional studies, the ensemble mean (the most typically used configuration of the data) is not ideal and some care needs to be given to assessing whether an individual ensemble member is a more appropriate representation for that region. It is our intention that this study is useful when making this determination in future.

Our results also indicate that regardless of the ensemble configuration, WetCHARTs performs poorly at reproducing the observed seasonal cycle in some regions. When no ensemble member is capable of reproducing the observed seasonal cycle

signal it suggests a deficiency in the parametrisations used. Understanding this behaviour is valuable as it can be used to identify processes that occur in a particular region that are not captured by a simple data-driven approach. This then informs the development of more complex land-surface models where such processes will need to be explicitly included. In order to address this, detailed analysis of some of the more challenging exemplar regions (Paraná River, Congo, Sudd and Yucatán) was performed.



For the Paraná River region in South America, a region we identified in Parker et al. (2018) as having the potential for significant CH$_4$ emissions driven by overbank inundation, we find that WetCHARTs typically reproduces the seasonality (r = 0.93) but underestimates the magnitude (-10.5 ppb). This underestimation is found to be most severe in specific years (2010, 2016, 2017) where the Paraná River and the Paraná Delta are flooded. This case highlights one deficiency with a data-driven
approach where the variability in wetland extent is forced by precipitation as in WetCHARTs, namely that the effects of significant river flow upstream of the wetland area (e.g. during a strong El Niño event) are not captured. In this instance WetCHARTs could act as a valuable benchmark against which to evaluate more complex land surface models which include lateral river flow and subsequent overbank inundation (Dadson et al., 2010).

For the Congo region in central Africa, previous studies performing flux inversions have suggested that the WetCHARTs
emissions are too high (Maasakkers et al., 2019; Lunt et al., 2019) but have not hypothesised on a cause. In this work we confirm that the magnitude of emissions from the WetCHARTs ensemble is inconsistent with atmospheric CH$_4$ observations with a much stronger rainy season wetland signal from WetCHARTs (20.5 and 23.6 ppb) than from observations (11.9 ppb) and that the observed seasonal cycle is poorly represented (r <= 0.67). We do however find that the spatial extent of the wetland emissions is largely in agreement with observations and that neither wetland extent parametrisation out-performs the other. Our
results point to the wetland fraction (Figure 12) being far too large compared to the JRC Surface Water Extent. When coupled with strong heterotrophic respiration from CARDAMOM, this results in excess emissions. This region highlights the importance (and uncertainty) of the underlying heterotrophic respiration and is a strong argument for the approach that WetCHARTs takes in its ensemble approach by utilising 9 heterotrophic respiration models in its full ensemble (FE) configuration. Utilising the same approach in the extend ensemble configuration (EE) used here (upon availability of suitable model data) would help
to further constrain the wetland emissions and be a useful addition to WetCHARTs.

The Sudd is the second region in central Africa that we focused on as it provided a stark contrast to the Congo. Whilst the Congo had significant variability with very strong emissions, we found that the Sudd region had very low emissions with very little variability in the WetCHARTs seasonal cycle which is inconsistent with the knowledge that this region is dominated by seasonally inundated wetlands. Indeed our satellite observations showed the strong seasonal cycle that was expected, pointing
to a deficiency in WetCHARTs in this region, leading to an extremely poor correlation (r = 0.2) to the observations and making this an interesting case study. Our investigation showed that the reason for the lack of seasonality in the WetCHARTs data was due to a strong anti-correlation between the respiration/temperature and the wetland extent. The seasonality of the wetland extent is in agreement with the observed CH$_4$ signal, both peaking during the August-November rainy season which leads us to conclude that lack of sufficient temperature/respiration is the reason for the overall lack of a strong CH$_4$ seasonal cycle. This
result again places WetCHARTs in the position to act as a useful benchmark when assessing these underlying fundamental processes within more complex land surface models.

The Yucatán region is our final region of focus. While the agreement between the emissions and observations is reasonable, WetCHARTs does underestimate the observed emissions during their peak each year. Furthermore, WetCHARTs produces a double-peak in the seasonality that is not present in the observations. We attribute both of these discrepancies to the wetland
extent parametrisations used. Both wetland extent parameters exhibit a double-peak, driven by the variability in precipitation





and by examining the spatial extent of the wetland datasets (Figure 15) we conclude that neither represent the large wetland complexes in this region, with GLWD doing particularly poorly. This result is of interest as the GLWD-based ensemble members are generally found to out-perform GlobCover for the majority of regions and overall we would conclude that GLWD provides a better representation of wetlands but that is not the case in this region. This highlights the ongoing need for further

improvements to global wetland extent datasets.

To conclude, we have performed the first, detailed, global and regional evaluation of the WetCHARTs $CH_4$ emission model ensemble against a long time series of high-quality, validated, satellite $CH_4$ observations. Our findings provide confidence that WetCHARTs is generally very capable of reproducing the observed wetland $CH_4$ seasonal cycle for the majority of wetland regions but that certain ensemble members are more suited to specific regions, either due to deficiencies in the underlying

data driving the model or complexities in representing the processes involved. The need for more reliable, validated, long-term wetland extent observations is clear as many of the discrepancies we observed are attributed to deficiencies in our knowledge of wetland extent. The remaining driving data (i.e. heterotrophic respiration and temperature) are shown to also contribute to the mismatch to observations, with the details differing on a region-by-region basis but generally showing that some degree of temperature dependency is better than none. Utilisation of an ensemble of heterotrophic respiration models for the full

WetCHARTs period would prove particularly valuable in this respect.

Finally, the data-driven approach utilised to produce WetCHARTs is well-suited to produce an ensemble dataset against which to evaluate more complex process-based land surface models that explicitly model the hydrological behaviour of these complex wetland regions.

*Data availability.* The latest version of the University of Leicester GOSAT Proxy v9.0 $XCH_4$ data (Parker and Boesch, 2020) is available

from the Centre for Environmental Data Analysis data repository at https://doi.org/10.5285/18ef8247f52a4cb6a14013f8235cc1eb. The version used in this work (v7.2) is available via contacting the author. WetCHARTs v1.0 is available from Bloom et al. (2017a). This study uses v1.2.1 which is available on request from A. Bloom.



## Appendix A: Appendix A - Regional Correlation Coefficients

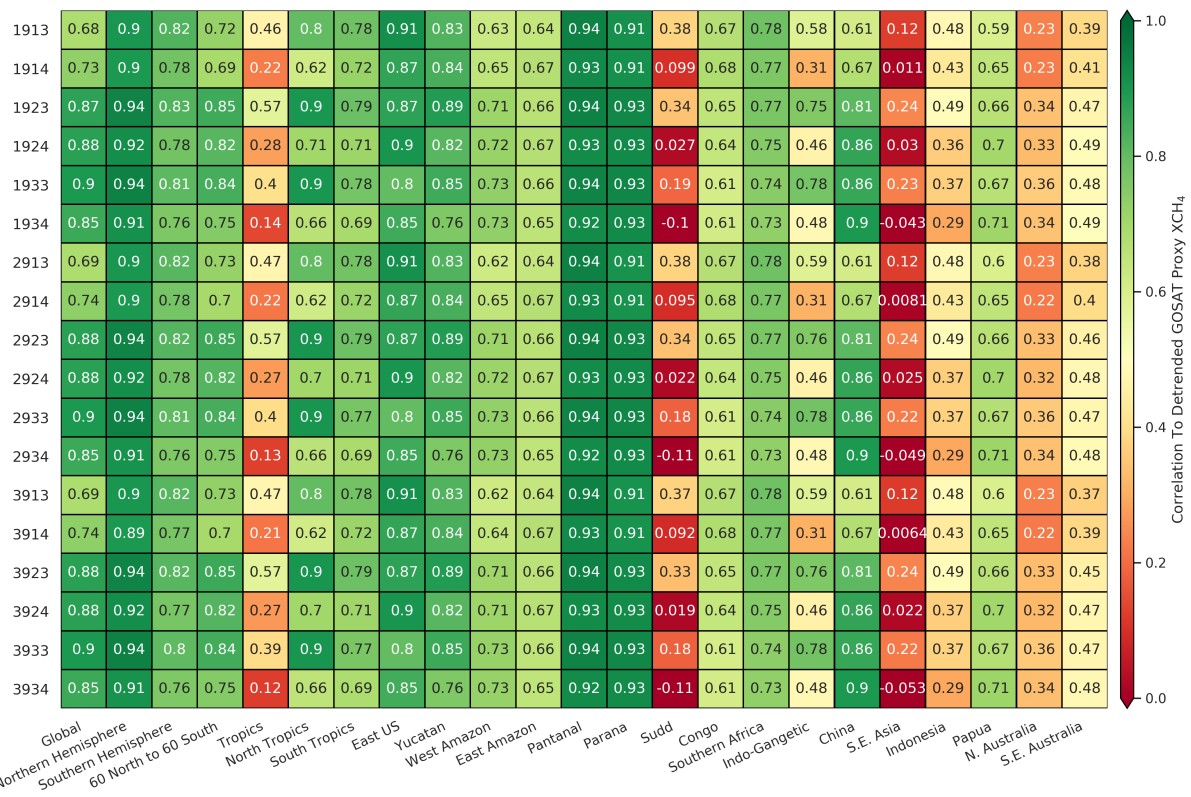

**Figure A1.** Table showing the correlation coefficients between the modelled and observed seasonal cycle for all regions for every ensemble member.

A detailed breakdown of the correlation coefficients between the modelled and observed seasonal cycle for all regions for all ensemble members is presented in Figure A1.

Several regions have a very poor correlation to the observations (Sudd, S.E. Asia, Indonesia, N. Australia and S.E. Aus-
5   tralia) across all ensemble members. It is also apparent that the temperature dependency is clearly significant for some regions, as demonstrated by the improved correlation coefficient for the **xxCx** ensemble members for West Amazon, China, N. Australia and S.E. Australia when comparing no temperature dependency (**xx1x**) against an increased temperature dependency (**xx2x/xx3x**). However, the temperature dependency seems to have little effect on other regions (East Amazon, East US, Yucatán, Pantanal and Paraná) and for some regions, the strongest correlation is found when there is no temperature dependency
10   and worsens when the temperature dependency is increased (e.g. Indonesia, Congo, Southern Africa).

The importance of the wetland extent parametrisation is region-specific, with the correlation to observations for the Tropics (especially North Tropics) being much better for GLWD extent (0.80-0.90 for North Tropics) vs GlobCover extent (0.62-





0.71). Some regions are largely unaffected by the extent parametrisation in terms of their seasonality (e.g. West Amazon, East Amazon, Pantanal, Paraná, Congo) whilst some regions are significantly affected (e.g. Sudd, Indo-Gangetic, S.E. Asia).





*Author contributions.* RJP generated the GOSAT XCH$_4$ retrievals, performed the analysis and wrote the manuscript. AAB produced an updated version of the WetCHARTs dataset for use in this study. CW and MPC produced the TOMCAT model simulations. All co-authors contributed to the planning and discussion of this study and on refining the manuscript.

*Competing interests.* We declare no knowledge of any competing interests.

*Acknowledgements.* RJP, HB, CW and MPC are funded via the UK National Centre for Earth Observation (NE/R016518/1 and NE/N018079/1). JM acknowledges financial support from the Horizon 2020 CHE Project (776186). We acknowledge the support of the UK Natural Environment Research Council through the grant The Global Methane Budget (MOYA, NE/N015681/1, NE/N015657/1 and NE/N015746/1). Part of this research was carried out at the Jet Propulsion Laboratory, California Institute of Technology, under a contract withthe National Aeronautics and Space Administration. Funding for the WetCHARTs emissions was provided through a NASA Carbon Monitoring System

Grant NNH14ZDA001N-CMS. We also acknowledge funding from the ESA GHG-CCI and Copernicus C3S projects.

We thank the Japanese Aerospace Exploration Agency, National Institute for Environmental Studies, and the Ministry of Environment for the GOSAT data and their continuous support as part of the Joint Research Agreement. This research used the ALICE High Performance Computing Facility at the University of Leicester for the GOSAT retrievals and analysis. The TOMCAT simulations were performed on the national Archer and Leeds Arc HPC systems.

The MODIS Surface Reflectance 8-Day L3 data and MODIS Combined 16-Day NDWI data were visualised via the Google EarthEngine software with the data provided courtesy of the NASA EOSDIS Land Processes Distributed Active Archive Center (LP DAAC), USGS/Earth Resources Observation and Science (EROS) Center, Sioux Falls, South Dakota (https://lpdaac.usgs.gov).



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
