# Peer review of "Exploring Constraints on a Wetland Methane Emission Ensemble (WetCHARTs) using GOSAT Satellite Observations"

_Biogeosciences, 2020_

## Referee Comment (RC1) · Anonymous Referee #1 · 26 Aug 2020

**General comments:**

The manuscript by Parker et al. provides an in-depth analysis of the correlation between seasonal variations in methane concentration based on the results of ensemble calculations using the WetCHARTs model and satellite-based observations on a global and regional scale. The set of methane emission data from wetlands is derived using external data on soil temperature, precipitation, heterotrophic respiration and wetland extent. The wetland emission data then processed together with anthropogenic and the other natural sources emission data using TOMCAT atmospheric chemistry transport model. The output data from TOMCAT simulations are compared with GOSAT
satellite observation data. The paper is well written and structured but needs some improvement and clarification.

Specific comments:

P3,4: Global scale factor on the fig.1 is obviously not the same thing with "s" in equation 1, although the first paragraph on page 4 says otherwise. The right equation for "s" is given by equation 3 in [Bloom et al., 2017b]. It takes its own value for each of the 18 members of the ensemble.

P3: "V1.2.1 of WetCHARTs has improved North American wetlands". This seems to need some explanation.

P3: I would also recommend to mention Eliseev et al. 2008 paper.

Section 2: No information on temperature data used for q10 dependence.

P4: Non-wetland ch4 emissions for TOMCAT are set using EDGAR (v4.2) data. Are such data available for the simulated period (2009-2017)?

Section 6: "The wetland extent is found to be the dominant explanation for the variance in all regions". Unfortunately, only one model of soil heterotrophic respiration (CAR-DAMOM) is used in this work. Based on formula 1 in this paper and fig. S2 in [Bloom 2016], the strong divergence between the data from different models, especially in the tropics, can significantly affect the variations in the seasonal cycle of methane. For some regions (especially S.E. Asia and Indonesia) low correlation may be partly due to the use of annually-repeating values for rice paddy emissions. They can be comparable or even exceed wetland emissions, have their own seasonal cycle, and are highly dependent on the same meteorological parameters (temperature and precipitation).

Technical comments:

P2, L25: "wetland ch4 seasonal cycle", which does make sense, transforms to P5, L25: "ch4 wetland seasonal cycle", which does not, and then just to "wetland seasonal
cycle" (seems incorrect) mostly used till the end of the paper. I would recommend to use the 1st sentence throughout the manuscript.

P8, L5: "observed emissions". I think here should be something like "variations in the WetCHARTs emissions"

P14, Fig.10 caption: ror

P23, L17: the sentence "argument for the approach that WetCHARTs takes in its ensemble approach" needs revision

---

## Referee Comment (RC2) · Anonymous Referee #2 · 7 Sep 2020

This manuscript presents an evaluation of the WetCHARTs product using GOSAT methane measurements taken over the 2009-2017 period. The TOMCAT model is used to represent these modelled measurements. This analyses both the seasonality and amplitude of the wetland methane emissions on global and regional scales. Wetland models of this type are of great value to the community and it is important that the impact of the drivers, limitations and applicability is understood. This paper addresses this by highlighting both the strengths and shortcomings of the model and undertakes a thorough analysis. It is well-suited for publication in this journal given the clarifications highlighted below.
[pg 4 Line 27] When using the EDGAR database - how are years beyond the range of the dataset represented (2012)? Has any trend being included to cover 2009-2017 period? This is also annual dataset so have any seasonal cycles associated with different regions been included?

[pg 4 Line 27] Similarly which data has been used for GFED 4.1s to cover 2009-2017? Has the beta release data been used for the latter years?

[pg 5 Lines 11-14] Are any corrections made to the GOSAT dataset based on the comparison to the TCCON network? If so, is this a global correction or are any regional variations taken into account?

[pg 5 Line 29-31] "This makes the assumption that wetlands dominate the uncertainty in interannual variability of the CH4 emissions and the remaining CH4 sources are in comparison far less uncertain" - is this a valid assumption for all regions? Could biomass burning have larger uncertainty in some regions?

[pg 6 Figure 2] Why is there are 1.00 correlation between ensemble members 1913,.. and 2913,.. (and 3913,..)? These different ensemble members represent those where the global scale factor has been altered - so is this expected? It would be worth clarifying this as the different ensemble members look very well correlated overall but this is perhaps a bit misleading if this is just due to a scaling factor difference in some cases.

[pg 7 Figure 3 / pg 9 Line 5-8] Are the defined East and West Amazon (and perhaps the Congo) regions, traversing the equator? How does their regional behaviour relate to the hemispheric differences seen? May be easier to see this if the equator/tropics were plotted in Figure 3.

[pg 8 Figure 4] What do the ranges here represent for the box and whisker plot? e.g. Is this presenting the mean or the median? Is the box representative of 1 sigma uncertainty or the inter-quartile range or something else? Please just clarify this within the caption (and does this match to Figure 6 and 7?).
[pg 10 Figure 5 & pg 14 Figure 10] "Colours indicate the different groupings" - which groupings do each of the colours represent? There seems to be different colours for panels 1 and 2 than panel 3 (e.g. pink)? Without additional clarification here this does not make it clear which ensembles show better correlation / lower standard deviation.

[Section 8, Case study 2: Congo] The implication here is that the low correlation with temperature dependence in the Congo could be due to the high magnitude. Since globally, and for most other regions, some temperature dependence is seen, would this explain the difference in this region?

[Section 9, Case Study 3: Sudd] Would be useful to clarify what the underlying temperature database being used is. Is there any reason why this would be misrepresenting the temperature variations in this region?

— Technical corrections:

[pg 9 Line 14] Commas should be added around "however" -> This, however, is not the case

[pg 14 Figure 10] "ror" -> "for"

BGD

---

## Author Comment (AC1) · 2 Oct 2020

**Response to RC1 Review Comments**

(Original Comment, Our Response, New Manuscript Text)

Firstly, we would like to express our gratitude to the editor and reviewers for providing a thorough review of our paper. We appreciate their efforts, especially in these difficult times.

**Specific comments:**

P3,4: Global scale factor on the fig.1 is obviously not the same thing with "s" in equation1, although the first paragraph on page 4 says otherwise. The right equation for "s" is given by equation 3 in [Bloom et al., 2017b]. It takes its own value for each of the 18members of the ensemble.

You are correct. In an attempt to be concise, we omitted this detail and accept that we need to be clearer in the text. We will modified the text to clarify this and reference the Bloom paper in more detail.

s is a model-specific scaling factor (Bloom et al., 2017), derived such that model annual emissions amount to either 124.5, 166 or 207.5 Tg/yr (see Figure 1 for model configuration details).

P3: "V1.2.1 of WetCHARTs has improved North American wetlands". This seems to need some explanation.

We will clarify by adding the following statement to the text:

WetCHARTs v1.2.1 wetland extent across Lehner & Doll (2004) wetland complex classes 0-25%, 25-50% and 50-100%, were scaled by 12.5%, 37.5% and 75%, respectively.

P3: I would also recommend to mention Eliseev et al. 2008 paper.

Noted, citation added.

Section 2: No information on temperature data used for q10 dependence.

We have clarified this in the text (and please see similar RC2 comment).

ERA-Interim skin temperature is used as the underlying temperature driving data.

P4: Non-wetland $CH_4$ emissions for TOMCAT are set using EDGAR (v4.2) data. Are such data available for the simulated period (2009-2017)?

The EDGARv4.2 database runs up to 2012, and we repeated the 2012 emissions for the remaining years, with no seasonal cycle applied. For the wetland regions in which

we're interested, any local seasonal cycle due to anthropogenic flux is likely very small compared to the natural sources, but we now note this possibility in the text.

The EDGARv4.2 database runs up to 2012, and we repeated the 2012 emissions for the remaining years, with no seasonal cycle applied. As we focus primarily over wetland emission areas, the local seasonal cycle due to anthropogenic fluxes is likely very small compared to these natural sources. We do however note the possibility that this effect could be a source of uncertainty.

Section 6: "The wetland extent is found to be the dominant explanation for the variance in all regions". Unfortunately, only one model of soil heterotrophic respiration (CARDAMOM) is used in this work. Based on formula 1 in this paper and fig. S2 in [Bloom2016], the strong divergence between the data from different models, especially in the tropics, can significantly affect the variations in the seasonal cycle of methane. For some regions (especially S.E. Asia and Indonesia) low correlation may be partly due to the use of annually-repeating values for rice paddy emissions. They can be comparable or even exceed wetland emissions, have their own seasonal cycle, and are highly dependent on the same meteorological parameters (temperature and precipitation).

We realise that we have not been clear enough in this section and we will rectify that in the revised manuscript. For clarity, this assessment of the variance is an assessment of the input driving WetCHARTS (Extent, Temperature, Respiration) vs the $CH_4$ emissions generated by WetCHARTs. It is not meant to be taken as a general statement about the importance of these parameters to explaining the variance in the real world. There are other factors that would need to be included to accomplish that. It is purely an assessment of how important these factors are and their influence on the resulting WetCHARTs modelled $CH_4$ fluxes. This assessment is useful as if a certain driver is dominating the response in WetCHARTs emissions and we subsequently observe discrepancies to the $CH_4$ measurements, it indicates further evaluation of that driving data may be useful in explaining these. We will add a statement similar to above to

clarify this in the manuscript.

To clarify, this analysis is purely an assessment of the WetCHARTs $CH_4$ emissions against its own driving data used to generate the emissions. It is not intended to be interpreted as a general statement about the importance of these parameters to explaining the variance in the real world. This assessment is useful as if a certain driver is dominating the response in WetCHARTs emissions and we subsequently observe discrepancies to the $CH_4$ measurements, it indicates further evaluation of that driving data may be useful in explaining these.

In addition, in specific relation to the comment above regarding soil heterotrophic respiration, we will add in a statement to explicitly acknowledge the under-representation of Rhet uncertainty in the discussion section.

Although for the extended period examined here we only have 1 heterotrophic respiration model available, the contribution of heterotrophic respiration uncertainty within the WetCHARTs Full Ensemble is considerable due to model disparities in mean emission rates and the corresponding seasonal cycles (see Figure 6 in Bloom et al., 2017, attached to this response as Figure 1). Ultimately further expansion and exploration of the heterotrophic respiration model ensemble may prove useful for robustly representing the terrestrial C cycling uncertainty.

**Technical comments:**

P2, L25: "wetland $CH_4$ seasonal cycle", which does make sense, transforms to P5,L25: "$CH_4$ wetland seasonal cycle", which does not, and then just to "wetland seasonal cycle" (seems incorrect) mostly used till the end of the paper. I would recommend to use the 1st sentence throughout the manuscript.

Noted and modified in the text.

P8, L5: "observed emissions". I think here should be something like "variations in the WetCHARTs emissions"

Noted and modified in the text.

P14, Fig.10 caption: ror

Noted and modified in the text.

P23, L17: the sentence "argument for the approach that WetCHARTs takes in its ensemble approach" needs revision

Noted and modified in the text.
* * *
[Figure]

**Fig. 1.** Figure 6 from Bloom et al., 2017 - The dominant uncertainty attribution in WetCHARTs

---

## Author Comment (AC2) · 2 Oct 2020

**Response to RC2 Review Comments**

(Original Comment, Our Response, New Manuscript Text)

Firstly, we would like to express our gratitude to the editor and reviewers for providing a thorough review of our paper. We appreciate their efforts, especially in these difficult times.

[pg 4 Line 27] When using the EDGAR database - how are years beyond the range of the dataset represented (2012)? Has any trend being included to cover 2009-2017 period? This is also annual dataset so have any seasonal cycles associated with different regions been included?

The EDGARv4.2 database runs up to 2012, and we repeated the 2012 emissions for the remaining years, with no seasonal cycle applied. For the wetland regions in which we're interested, any local seasonal cycle due to anthropogenic flux is likely very small compared to the natural sources, but we now note this possibility in the text.

The EDGARv4.2 database runs up to 2012, and we repeated the 2012 emissions for the remaining years, with no seasonal cycle applied. As we focus primarily over wetland emission areas, the local seasonal cycle due to anthropogenic fluxes is likely very small compared to these natural sources. We do however note the possibility that this effect could be a source of uncertainty.

[pg 4 Line 27] Similarly which data has been used for GFED 4.1s to cover 2009-2017? Has the beta release data been used for the latter years?

We used the GFEDv4.2 emissions for the correct year up and including to 2016, and used a climatology for 2017 and 2018. We did not use the beta release due to availiability when this study commenced. This may affect the modelled seasonal cycle in some regions for the latter two years but will not affect any conclusions.

We used the GFEDv4.2 emissions for the correct year up and including to 2016, and used a climatology for 2017 and 2018.

[pg 5 Lines 11-14] Are any corrections made to the GOSAT dataset based on the comparison to the TCCON network? If so, is this a global correction or are any regional variations taken into account?

We will include this statement to clarify this: After performing extensive validation to TCCON, we subtract one global offset from the GOSAT data. This value is typically small. For v7.2 of the data (as used in this study, the value was 7.71 ppb). For our latest data, v9.0, the value is 9.06 ppb (see Parker et al, 2020, under review)

[pg 5 Line 29-31] "This makes the assumption that wetlands dominate the uncertainty in interannual variability of the $CH_4$ emissions and the remaining $CH_4$ sources are in comparison far less uncertain" - is this a valid assumption for all regions? Could biomass burning have larger uncertainty in some regions?

This is a good question, and whilst it is generally true that the uncertainty regarding wetland $CH_4$ flux is much larger than that of biomass burning, it could be the case in some regions that the local biomass burning uncertainty is not insignificant. We have carried out full global inversions of $CH_4$ flux using this GOSAT data product and this chemical transport model (paper in prep). Whilst it is difficult to separate out the fire emissions from other emissions sectors using this methodology, our findings suggest that flux changes in burning regions in South America and Africa during the burning season generally change relatively little from the prior, compared to nearby wetland regions. However, it is true that in some extreme years (e.g. 2010 drought in S. America), there are more significant changes to the GFED prior derived by the inversion. Although wetland and burning regions are often spatially distinct, this could affect some of our results to a small extent and we have highlighted this possibility in the main text.

It should be noted that there is the potential for our assumptions regarding biomass burning emissions to interfere with our derived wetland seasonal cycle. However, we have carried out full global inversions of $CH_4$ flux using this GOSAT data product and this chemical transport model (paper in prep) which suggest that this is not a significant issue. Whilst it is difficult to separate out the fire emissions from other emissions sectors using this methodology, our findings suggest that flux changes in burning regions in South America and Africa during the burning season generally change relatively little from the prior, compared to nearby wetland regions. However, it is true that in some extreme years (e.g. 2010 drought in S. America), there are more significant changes to the GFED prior derived by the inversion. Although wetland and burning regions are often spatially distinct, this could affect some of our results to a small extent.

[pg 6 Figure 2] Why is there are 1.00 correlation between ensemble members

1913,..and 2913,.. (and 3913,..)? These different ensemble members represent those where the global scale factor has been altered - so is this expected? It would be worth clarifying this as the different ensemble members look very well correlated overall but this is do only perhaps a bit misleading if this is just due to a scaling factor difference in some cases.

You are correct in that these ensemble members (e.g. 1913, 2913, 3913) do only differ by a scale factor. We will add a statement in the text to highlight this to the reader so that it is clear. It should also be noted that a change to the global scale factor does not necessarily just result in a completely linear change in the atmospheric concentrations (e.g. due to non-linearity in transport, OH sink, etc). We do think, regardless of this, this plot is informative and provides useful information on the inter-relation between ensemble members. It also provides justification for taking representative ensemble members in some of the latter analysis.

It should be noted that the high correlation between certain groups of ensemble members is expected (e.g. for members 1913, 2913, 3913 where the only configuration difference is the global scaling factor).

[pg 7 Figure 3 / pg 9 Line 5-8] Are the defined East and West Amazon (and perhaps the Congo) regions, traversing the equator? How does their regional behaviour relate to the hemispheric differences seen? May be easier to see this if the equator/tropics were plotted in Figure 3.

This is a good point. We will add the equator line on to Fig 3 for clarity. When examining these regions which do (slightly) cross the equator we see no significant North-South gradient on the scale of the region.

[pg 8 Figure 4] What do the ranges here represent for the box and whisker plot? e.g. Is this presenting the mean or the median? Is the box representative of 1 sigma uncertainty or the inter-quartile range or something else? Please just clarify this within the caption (and does this match to Figure 6 and 7?)

We will clarify this in the text. It is the median and the 25th/75th percentiles.

The median and 25th/75th percentiles are indicated.

[pg 10 Figure 5  pg 14 Figure 10] "Colours indicate the different groupings" – which groupings do each of the colours represent? There seems to be different colours for panels 1 and 2 than panel 3 (e.g. pink)? Without additional clarification here this does not make it clear which ensembles show better correlation / lower standard deviation.

We will endeavour to explain this better in the text as we accept this style of figure requires more explanation. As we are holding two parameters fixed and only plotting the data associated to a change in the third parameter, the number of joined data points will depend on how many dimensions that parameter space has. As an example, we have 2 wetland extent maps, so 9 out of the 18 total ensemble members for each. This results in 9 sets of lines for the 3rd panel. In comparison, the other two parameters have 3 variables resulting in 6 ensemble members each and hence 6 sets of lines in the first two panels. The colours of these lines are arbitrary and only really intended to distinguish the lines from each other within a panel. There is no link between colours in different panels and we will make this clear in the text. We considered making this figure black/white but as some lines closely overlap each other, we felt it was useful to be able to see the separation of lines by colouring them.

The line colours on these plots are to indicate the different combinations within each panel. Due to the nature of the plot, there is no link between colours in different panels as they represent different pairs/trios of data.

[Section 8, Case study 2: Congo] The implication here is that the low correlation with temperature dependence in the Congo could be due to the high magnitude. Since globally, and for most other regions, some temperature dependence is seen, would this explain the difference in this region?

This is difficult to give a definitive answer and future studies focused specifically on the

Congo region may help. We will add the following statement in the manuscript:

Various published atmospheric inversions of our $CH_4$ data that have used WetCHARTs as the prior, all indicate that the Congo emissions are over-estimated by WetCHARTs and reduced when confronted with observations. This highlights the large uncertainty over this region. Once the necessary MsTMIP (or similar) model data becomes available and it is possible to extend the WetCHARTs Full-Ensemble (i.e. all respiration models) to this time period we will revisit this question in a future study.

[Section 9, Case Study 3: Sudd] Would be useful to clarify what the underlying temperature database being used is. Is there any reason why this would be misrepresenting the temperature variations in this region?

The data used to drive WetCHARTs is as described as in Bloom et al., 2017 (https://doi.org/10.5194/gmd-10-2141-2017). The temperature is ERA-Interim skin temperature. Again, once a more complete Full-Ensemble of WetCHARTs is available, it should allow a future study to further disentangle the effects of these different parameters.

ERA-Interim skin temperature is used as the underlying temperature driving data. One limitation of skin temperature is the assumption that heterotrophic respiration is sensitive to top of soil temperature. We advocate for an expansion of the WetCHARTs ensemble to include subsurface soil temperature estimates - in place of surface skin temperatures - to explicitly represent the representation uncertainty associated with the soil temperature dependency of methanogenesis.

**Technical corrections:**

[pg 9 Line 14] Commas should be added around "however" –> This, however, is not the case

Noted and modified in the text.

[pg 14 Figure 10] "ror" –> "for"

Noted and modified in the text.